# Veined Rock Performance under Uniaxial and Triaxial Compression Using Calibrated Finite Element Numerical Models

**Gisèle A. Rudderham and Jennifer J. Day \***

Department of Geological Sciences and Geological Engineering, Queen's University,
Kingston, ON K7L 3N6, Canada; g.rudderham@queensu.ca
\* Correspondence: day.jennifer@queensu.ca

**Abstract:** Geotechnical rockmass characterization is a key task for design of underground and open pit excavations. Hydrothermal veins influence excavation performance by contributing to stress-driven rockmass failure. This study investigates the effects of vein orientation and thickness on stiffness and peak strength of laboratory scale specimens under uniaxial and triaxial compression using finite element numerical experiments of sulfide veined mafic igneous complex (CMET) rocks from El Teniente mine, Chile. The initial numerical models are calibrated to and validated against physical laboratory test data using a multi-step calibration procedure, first of the unveined Lac du Bonnet granite to define the model configuration, and second of unveined and veined CMET. Once calibrated, the numerical experiment involves varying the vein geometry in the veined CMET models by orientation (5 to 85°) and thickness (1, 4, 8 mm). This approach enables systematic investigation of any vein geometry without limitations of physical specimen availability or complexity of physical materials. This methodology greatly improves the value of physical laboratory test data with a limited scope of vein characteristics by using calibrated numerical models to investigate the effects of any other vein geometry. In this study, vein orientation and thickness were both found to have a significant impact on the specimen stiffness and peak strength.

**Keywords:** triaxial compressive stress experiments; finite element method; geomechanics numerical modelling; numerical calibration; model validation; hydrothermal veined rocks; vein geometry

## 1. Introduction

Characterizing rockmass behaviour has become more important yet challenging as growing demand for underground excavation projects forces their construction into deeper and more geologically complex materials. Problematic rockmass behaviours such as rockbursts pose risk to human welfare and equipment condition and must be mitigated by adequately predicting rockmass instability behaviour and implementing appropriate risk reduction measures. In rockmasses with complex geometries of geological structures, the interaction between structures and in situ stresses can lead to stress-induced, structure-controlled failure due to stress concentrations that exceed brittle damage thresholds and strength [1,2]. The role of joints and other fractures (i.e., interblock rockmass structures per [3]) are understood by rock engineering practitioners to be a significant control on ground behaviour and are normally included in rockmass characterization (e.g., [4,5]). However, intrablock rockmass structures found in complex rockmasses, such as hydrothermal veins, are conventionally overlooked [3]. The strength of hydrothermal veins relative to the surrounding intact rock block (i.e., wallrock) is conventionally perceived to be high enough to be inconsequential to ground performance. This perception originates from a time when excavations were constructed at shallow depths and with simple geometries and geologies, such that gravity-driven rockmass failure dominated through interblock structures. Deep levels at El Teniente block caving mine in Chile have experienced localized

stress-driven, structure-controlled failure along veins [2,6]. Furthermore, Clark and Day [7] demonstrated in uniaxial compressive strength (UCS) laboratory tests on veined rocks that veins of different mineralogies and surrounded by different wallrock lithologies can have weakening or strengthening impacts on the measured geomechanical properties of the whole specimen. In massive, intact, veined rockmasses, the influence of veins on rock performance is amplified when joints are relatively sparse [8]. Brzovic and Villaescusa [6] demonstrated the importance of considering healed veins as discontinuities within a rockmass at El Teniente mine.

Numerical engineering design of excavations in rock requires quantification of geomechanical properties of intact rock, natural fractures, and hydrothermal veins, among others. The behaviours of these materials under load are simulated by assigning each type constitutive model parameters. Incorporating rockmass structures in numerical analysis of rock can be done using various approaches, ranging from continuum to discontinuum model elements. Using continuum materials, rockmass characterization schemes can be employed to account for discontinuities including veins and fractures such as the composite geological strength index [3] coupled with the generalized Hoek–Brown shear strength criterion [9,10]. In pseudo-discontinuum and discontinuum modelling methods, discrete elements representing individual geological discontinuities can be incorporated [8,11,12]. In finite element method (FEM) geomechanics numerical modelling software, pseudo-discrete joint elements enable simulation of discontinuities using macro-mechanical parameters that can be measured through physical laboratory testing. These parameters include elastic normal stiffness and shear stiffness [13], as well as peak and post-peak shear strengths through criteria such as Mohr–Coulomb [14,15] or Barton–Bandis [16]. The ability to simulate discontinuities as pseudo-discontinuum joint elements whose behaviour is defined by macro-mechanical parameters was the primary factor in selecting RS2, the 2D FEM software by Rocscience [17], for use in this study.

Geomechanics laboratory testing is a typical method to measure deformation and strength properties of discontinuities. These values are then utilized as inputs to numerical models of excavations in which joint elements are distributed spatially based on rockmass characterization data and the experience of the practitioner. However, there are often practical limitations to sample collection for physical laboratory testing campaigns, such as project budget or availability of samples with the desired characteristics, that inhibit detailed studies. This problem is addressed in this study by creating calibrated numerical models of laboratory-scale UCS and triaxial compressive strength (TCS) geomechanical experiments to investigate the influence of a single hydrothermal vein in a rock specimen for a broad range of vein geometries. TCS tests have been utilized by several researchers to measure the shear strength of intact rock structures [8,18,19]. This method may be preferred over direct shear tests, as no encapsulation material is required, sample preparation is simpler, and the load capacity of triaxial equipment is usually significantly higher than direct shear equipment, which is necessary to break through intact veins [20].

In this study, the model configuration and geomechanical input parameters are calibrated such that the resulting elastic responses and peak compressive strengths of the modelled specimens are equivalent to those reported by physical laboratory test results. The laboratory test results used for calibration are of (i) Lac du Bonnet (LdB) granite UCS and TCS specimens reported by Martin [21] and Labeid [22], and (ii) hydrothermally veined intact rock UCS and TCS specimens from El Teniente mine in Chile reported by de los Santos Valderrama [23]. The calibrated models are then utilized in a numerical TCS experimental suite to investigate the resultant geomechanical behaviour of single-veined numerically simulated specimens for a variety of geomechanical inputs and vein geometries.

## 2. Calibration Methodology

Several parts of the FEM model configuration can influence the results, including boundary and initial conditions, discretization and mesh geometry, platen–specimen contact joint element properties, and rate of applied displacement used to simulate application

of load in UCS and TCS experiments. The numerical calibration in this study involved varying these model settings so that the model output agreed with physical laboratory test data reported by Martin [21] and Labeid [22] of the LdB granite, which is a polycrystalline, homogeneous, low-porosity rock without veins. The LdB granite was chosen for this study as its detailed geomechanical properties are reported in the literature, and it exhibits low geomechanical variability. The reported properties provide clear targets for calibration of the FEM model configuration in this study, which, after calibration, may be used to simulate UCS and TCS experiments for any rock type. The calibrated model configuration was then used to simulate UCS and TCS tests of a heterogeneous veined rock, the Complejo Máfico El Teniente (CMET) unit from El Teniente mine, which is a suite of hydrothermally altered and veined mafic volcanic rocks. The physical laboratory test data reported by de los Santos Valderrama [23] were the most detailed found by the authors of this type of laboratory data at the time of this study and therefore chosen as the calibration target in this study for rock specimens each with a single inclined vein. The multi-step calibration methodology employed in this study is illustrated in Figure 1 and explained here:

1. Unveined UCS and TCS numerical calibration and validation of the FEM model configuration using LdB granite:

    a. Model configuration: external boundary; material boundaries defining dimensions of specimen and loading platens; position of x- and y-pinned and x- or y-restrained external and material boundary nodes and elements are defined to simulate physical laboratory conditions.

    b. Sensitivity analysis to model settings: Mesh element density, displacement rate, and stiffness properties of the platen–specimen joint element are varied.

    c. Comparison of numerical results to physical laboratory test data: Model settings are selected based on their relative influence on the model and whether the numeric output matches target output.

2. Unveined and veined UCS calibration and TCS validation using CMET mafic wallrock with single sulfide veins:

    a. Calibration of unveined specimen: Strength and stiffness properties of laboratory UCS test data compared to UCS model results.

    b. Incorporation of vein: Vein is incorporated into UCS model, first represented only by the intact vein material, followed by including joint elements at vein–wallrock contacts.

    c. Sensitivity analysis of inputs: Material properties of all components of the UCS models defined including wallrock and vein materials, and joint elements at vein–wallrock contacts. The solution is indeterminate, so each parameter cannot be individually calibrated. Therefore, a sensitivity analysis is employed to evaluate the effect of each unknown variable on peak strength of the specimen.

    d. Calibration of veined models to laboratory results: Inputs are adjusted based on the sensitivity analysis results to achieve a set of calibrated input parameters that produce UCS peak strength results that match the physical laboratory test data.

    e. Sensitivity analysis of model parameters on outputs: Calibrated parameters are systematically varied to ensure one parameter alone is not heavily influencing numeric output.

    f. Validation: TCS models are used to validate the output Mohr–Coulomb strength parameters against reported laboratory test data.

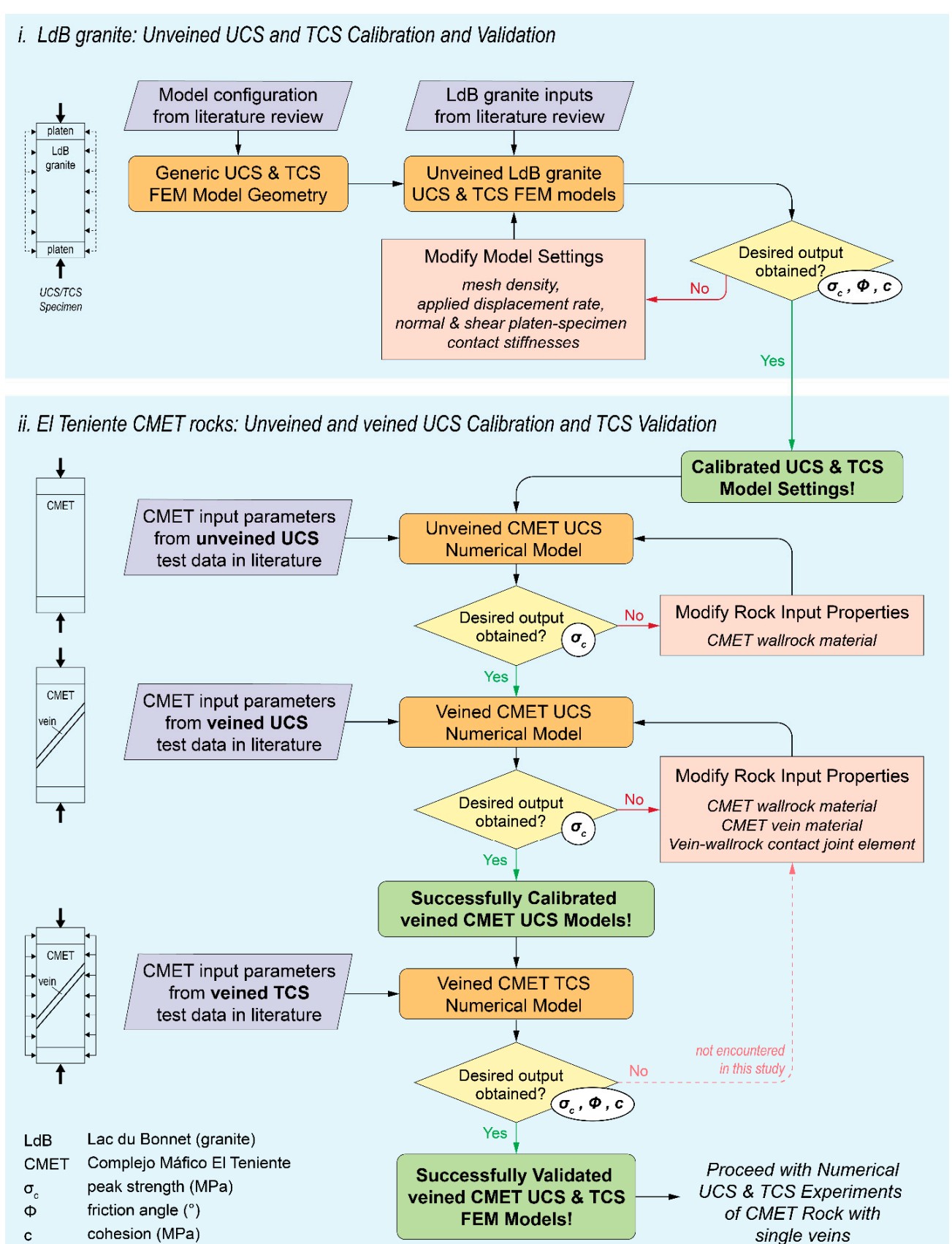

**Figure 1.** Workflow of procedures developed in this study to calibrate and validate 2D FEM numerical models of UCS and TCS experiments of rock specimens without and with one vein.

## 3. Model Configuration

The external boundary of the plane strain 2D FEM model used to simulate UCS and TCS tests measures 61 mm wide and 168 mm tall, which agree with the Labeid [22] physical laboratory test specimens and follows the ISRM [24] and Bieniawski and Bernede [25] suggested methods. Top and bottom platens are included in the external boundary dimension of the model, each with a height of 20 mm; thus, the specimen height is 128 mm and the height-to-diameter ratio is 2.10. The material input properties of the steel platens are listed in Table 1. Vertical downward displacement is applied to the specimen at the top edge of the external boundary through a staged displacement perpendicular to the external boundary that increases incrementally in every model stage. The displacement is applied using stage factors such that the first stage has a stage factor of zero, and with each subsequent stage this factor increases by one. As these are FEM models, each stage is an implicit solution to achieve equilibrium of the material based on principles of infinitesimal strain; as a result, there are no explicit time steps used to solve the computation [26]. A displacement-controlled configuration was utilized instead of load control because it more accurately reflects the physical laboratory test procedure used by Labeid [22] for specimens that are used as calibration targets in this study. Examples of these model configurations as well as external boundary conditions such as pins (x-displacement = y-displacement = zero) and rollers (x-displacement or y-displacement = zero) are illustrated in Figure 2.

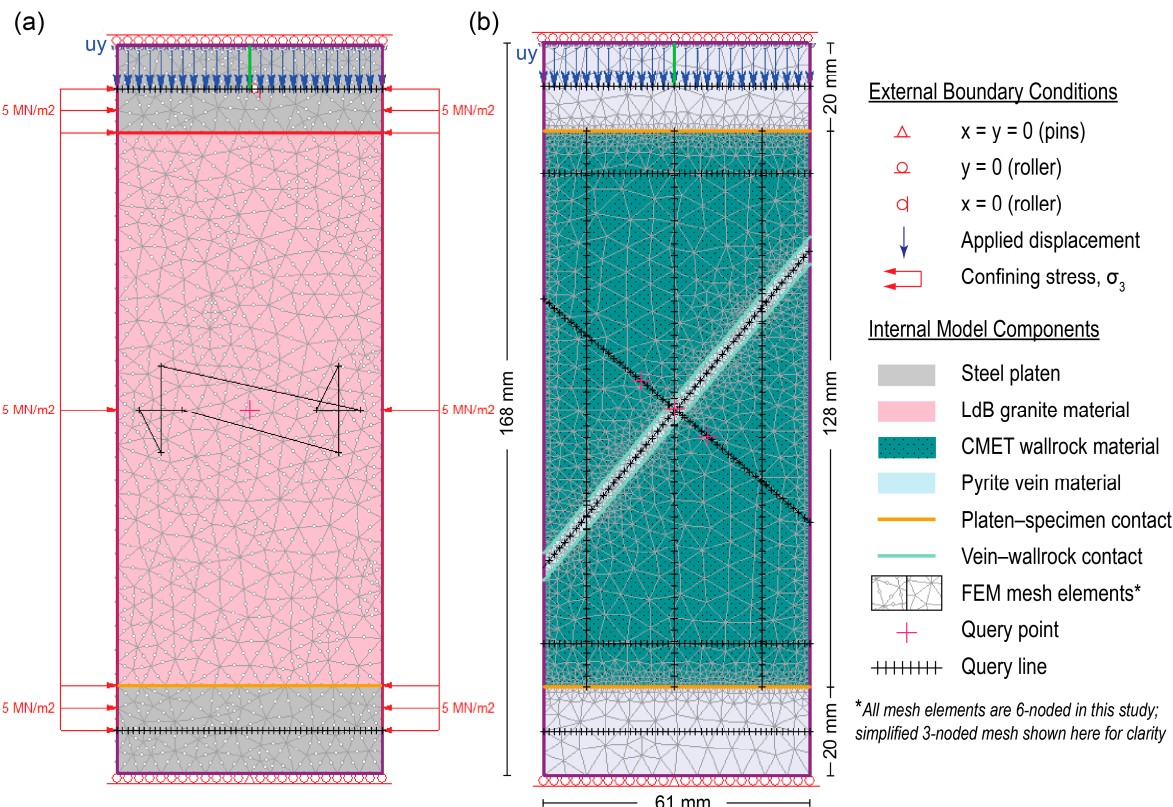

**Figure 2.** General 2D FEM model configurations used in this study of (**a**) LdB granite TCS specimen at $\sigma_3$ = 5 MPa and (**b**) veined CMET UCS specimen.

**Table 1.** Steel properties used as material inputs for FEM modelled platens (data from [27]).

| Parameter (Units) | Value |
|---|---|
| Unit weight, $\gamma$ (MN/m$^3$) | 0.077 |
| Young's modulus, E (GPa) | 200 |
| Poisson's ratio, $\nu$ | 0.27 |

Three sets of query locations to measure model outputs are used (Figure 2). Axial stress is measured horizontally at the centre of both platens. A query with 50 evenly spaced nodes is used across the width of each platen and averaged to calculate axial stress. Poisson's ratio ($\nu$) is calculated by recording strain measurements at 10 points on both the left and right sides of the specimen, in a cross shape to record both x- and y-directional strains. Young's modulus (E) is calculated for the vein (when present), the wallrock, and the whole specimen. For veined models, E is calculated in the vein at three points (centre and 2 cm in either direction along the vein) and the average is calculated. For the wallrock, E is calculated 2 cm perpendicular from the centre of the vein. For the whole specimen, E is calculated by querying axial strain across the width of the specimen, with query lines 1 cm below the top and 1 cm above the bottom of the specimen. The peak strength is then used to calculate E. For veined specimen models, the position of the vein is such that the centre of the vein is at the centre of the model.

## 4. Calibration of Lac du Bonnet Granite Models

The three major components of the UCS and TCS numerical model configuration that require calibration are (i) model geometry, (ii) applied displacement rate and mesh density, and (iii) properties of the platen–specimen contact joint element. Comparing the model outputs to published physical test data is used to validate the calibration. An iterative process was used whereby the first sensitivity analysis was completed, then numerical inputs were calibrated to laboratory test data of LdB granite published by Martin [21] and Labeid [22], and lastly the second sensitivity analysis was completed using the calibrated inputs.

The stiffness and strength properties for LdB granite used in this study are listed in Table 2. These published material properties are based on physical tests completed on 3D cylindrical specimens. To convert 3D Young's modulus and Poisson's ratio measured from physical laboratory test data ($E_{3D}$ and $\nu_{3D}$, respectively) to their 2D elastic plane strain equivalents for use as numeric inputs in this study ($E_{2D}$ and $\nu_{2D}$, respectively), Equations (1) and (2) [28,29] were used. Measurements of peak strength in the 2D plane strain numerical models are assumed to have no difference between a 3D physical test, based on [30].

$$\nu_{2D} = \frac{\nu_{3D}}{1 + \nu_{3D}} \tag{1}$$

$$E_{2D} = \frac{E_{3D}}{(1 - \nu_{2D}^2)} \tag{2}$$

### 4.1. Definition of Calibration Target

Influences of the mesh density and applied displacement rate in elastoplastic UCS models of LdB granite were evaluated by comparing the model output peak compressive strength ($\sigma_c$) to the analytical solution for the failure criterion used in the model, following guidance from Markus [29]. For the Mohr–Coulomb strength criterion, $\sigma_c$ is defined by Equation (3), where c is cohesion (MPa) and $\phi_i$ is internal friction angle (°). This calculated solution using the inputs from Table 2 give a $\sigma_c$ value of 216 MPa, which agrees with the physical test data by Labeid [22] and Martin [21] and is therefore used as the calibration target for mesh density and applied displacement rate.

$$\sigma_c = 2c \times \frac{\cos \phi_i}{1 - \sin \phi_i} \tag{3}$$

### 4.2. Calibration of Mesh Density

Mesh density controls the resolution of numeric outputs in an FEM model. Shorter computation times are desirable, and therefore a balance between model resolution and computational efficiency is needed. Mesh in RS2 software is generated by discretizing the external boundary and creating a mesh with a user defined number of mesh elements [17].

In the unveined rock models in this study, a uniform mesh distribution is used. In veined rock models, a graded mesh geometry is used to increase mesh density near the vein–wallrock contacts.

**Table 2.** Geomechanical properties of Lac du Bonnet granite.

| Parameter (Units) | Physical Laboratory Test Data | Numerical Input Value | Calibrated Numerical Output |
|---|---|---|---|
| Young's modulus, E (GPa) | 69.54 *; 69 ± 5.8 † | 64.83 § | 69.4 |
| Poisson's ratio, ν | 0.36 *; 0.26 ± 0.04 † | 0.26 § | 0.35 |
| Peak uniaxial compressive strength, $\sigma_c$ (MPa) | 228 *; 200 ± 22 † | - | 228.5 |
| Tensile strength, $\sigma_t$ (MPa) | 9.3 ± 1.3 † | 9.3 | 8.9 ** |
| Cohesion, c (MPa) | 30 †,‡ | 30.4 | 29.0 |
| Friction angle, ϕ (°) | 59 †,‡ | 62.2 | 61.6 |
| Unit weight, γ (MN/m³) | N/A | 0.027 | - |

Referenced data from: * [22], † [21], ** [20], ‡ [31], § calculated from Equations (1) and (2).

The sensitivity analyses of both uniform and graded mesh density on LdB specimen models in this study tested models containing between 400 and 19,000 nodes. The overall peak strength result is not significantly influenced by uniform or graded mesh density, as shown in Figures 3a and 3b, respectively. For all the remaining models in this study, a calibrated mesh density of 800 mesh elements or 1800 nodes is utilized, giving a nominal mesh element area of 12.8 mm²/element.

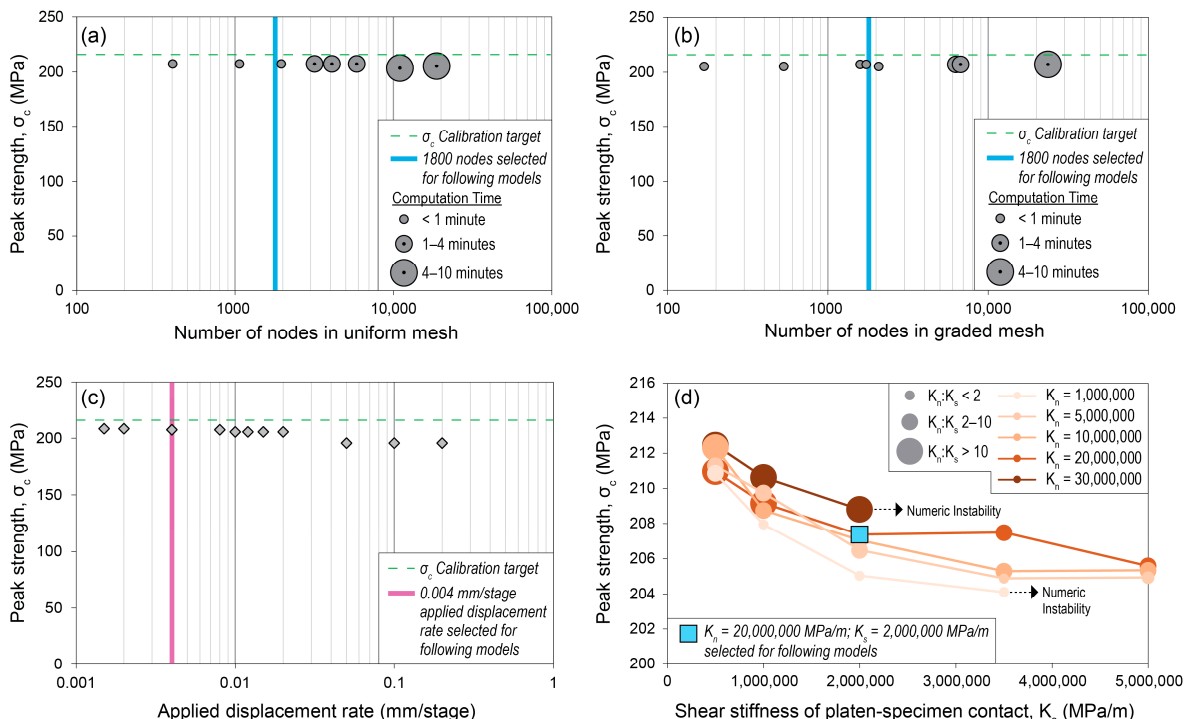

**Figure 3.** Model configurations of FEM LdB granite UCS models for (**a**) uniform mesh density, (**b**) graded mesh density, (**c**) applied displacement rate, and (**d**) normal stiffness ($K_n$) and shear stiffness ($K_s$) of platen–specimen joint element contact.

### 4.3. Calibration of Applied Displacement Rate

In a UCS or TCS test with a high displacement rate, the resolution of stress evolution is insufficient. In this case, large stress increments are applied to the specimen, leading to a premature interpretation of peak strength and onset of yield because the stage preceding yield is less than one stress increment from the true yield stress of the model. To determine the optimum applied displacement rate, a sensitivity analysis on LdB granite models was conducted to vary the applied displacement rates between 0.0015 and 0.2 mm/stage. Greater displacement rates lead to reduced peak strengths, as shown in Figure 3c. Based on the result of this sensitivity analysis, the optimal applied displacement rate is 0.004 mm/stage, which is utilized for the remainder of this study.

### 4.4. Stiffness and Strength of Platen-Specimen Interface

The platen–specimen interface in RS2 software can be defined by a material boundary or joint element. Material boundaries in RS2 separate regions of different material properties, but the boundary itself does not have definable geomechanical parameters. Joint elements are interfaces that allow movement as defined by input stiffness and strength properties [17]. A joint element most realistically replicates a laboratory UCS and TCS test setup because there is physically an open contact between the steel platens and a rock specimen.

Published data on stiffness and strength values for the platen–specimen interface are not readily available. For this study, the Mohr–Coulomb shear strength criterion was selected to define joint element strength. Cohesion and tensile strength were set to zero and friction angle ($\phi$) was calculated based on the coefficient of friction ($\mu$) between the steel platen and the rock specimen using Equation (4):

$$\phi = \tan^{-1}\mu \tag{4}$$

The coefficient of friction ($\mu$) between steel and rock used to calculate the friction angle in this study is 0.57, which represents the static friction between a dry interface of cast concrete and a steel plate [32]. Using Equation (4), the calculated friction angle used for the platen–specimen contact joint element is 30°. Surface degradation is not expected to occur as a result of shear displacement; therefore, residual friction angle is input as equal to peak friction angle.

Normal stiffness ($K_n$) and shear stiffness ($K_s$) properties of the platen–specimen interface were initially constrained by values published by Read and Stacey [33] and calibrated based a sensitivity analysis. The $K_n$ of rock discontinuities depends on the properties of the surrounding (wallrock) materials, matching (or mating) between the two surfaces, infill thickness and properties if present, and magnitude of normal stress increments [13]. Geological joints typically have a $K_n$:$K_s$ ratio of between 2 and 10 [33] and this ratio may tend toward 1 for intact veins [34]; therefore, a higher ratio is expected to represent the relatively low shear stiffness of a planar, smooth contact between the platen and specimen. Normal stiffness may increase as applied normal stress increases [35] and may be defined by a hyperbolic relationship [13].

In this sensitivity analysis of the platen–specimen interface on LdB granite models, normal stiffness values between 1,000,000 and 30,000,000 MPa/m and shear stiffness values between 500,000 and 5,000,000 MPa/m were tested for the platen–specimen contact joint elements. The resulting peak strength ($\sigma_c$) values in the UCS models of LdB granite decreased with increasing shear stiffness and slightly increased with increasing normal stiffness; $\sigma_c$ responses for all sensitivity analysis models are presented in Figure 3d. The calibrated platen–specimen interface stiffness values used for the remainder of this study are $K_n$ = 20,000,000 MPa/m and $K_s$ = 2,000,000 MPa/m. The stiffness between the platen and specimen should have a negligible effect compared to the stiffness of the material as displacement measurements during testing are accommodated by deformation of the specimen.

### 4.5. Validation of LdB Granite Models

To validate the calibrated model configuration settings and LdB granite input properties, quantitative UCS model outputs of elastic properties (E and ν) and peak strength (σ$_c$) were compared to published test data. Four FEM models were created to conduct this comparison: 1 UCS model and 3 TCS models (σ$_3$ = 5, 10, and 15 MPa) to obtain the strength envelope. The numerical output values are listed in Table 2 and the individual validation model results are summarized in Figure 4a. As shown in Figure 4b, the 2D elastic and peak strength outputs are well aligned with UCS test data from Labeid [22].

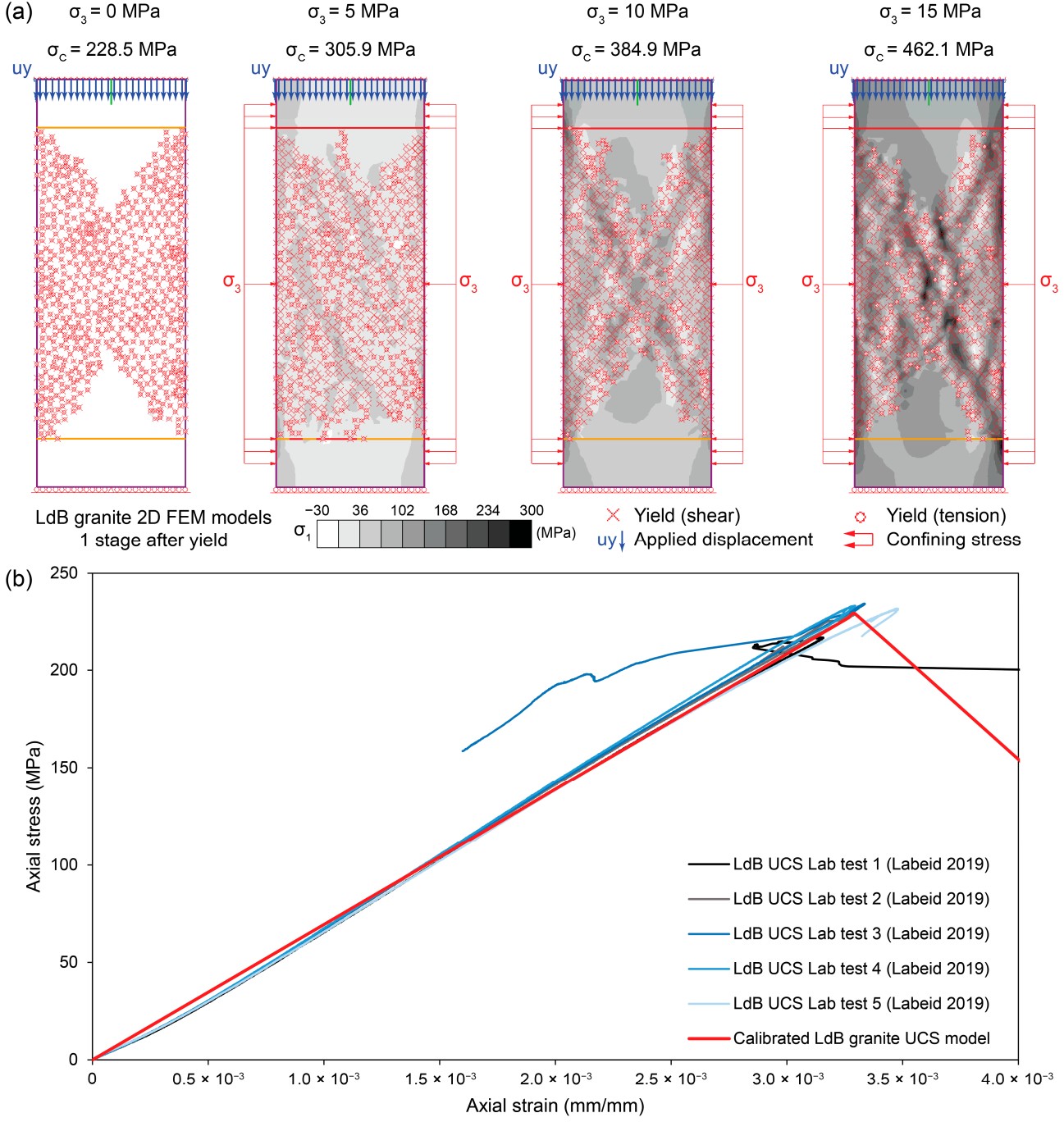

**Figure 4.** Results of LdB granite UCS and TCS model validation; (**a**) peak strengths of each model accompanied by images of post-yield major principal stress (σ$_1$) and yielded elements; (**b**) comparison of UCS numerical model result compared to laboratory test data from Labeid [22].

## 5. Calibration of CMET Models

Laboratory UCS and TCS test results published by de los Santos Valderrama [23] on specimens from El Teniente mine include geomechanical properties and geological descriptions of the lithologies, vein mineralogies, and vein geometries. The calibration workflow in this section systematically varies input parameters to obtain desired outputs. One vein geometry was modelled in the calibration model to reflect the laboratory test specimen calibration targets; specifically, with a vein thickness of 4 mm and a vein orientation relative to the core axis ($\alpha$) of 26°. Additional details about the calibration targets are discussed in Sections 5.1 and 5.2. The calibration methodology is summarized in Section 2 and Figure 1.

### 5.1. Numerical Input Parameters

The UCS and TCS models with single veins in this study include a vein material and two joint elements that represent the vein–wallrock contacts. The constitutive behaviour of these joint elements is defined by normal and shear stiffness ($K_n$ and $K_s$), and the Mohr–Coulomb shear strength criterion with cohesion (c), friction angle ($\phi$), and tensile strength ($\sigma_t$). Following guidance on Mohr–Coulomb strength of intrablock structures from Day et al. [34,35], $K_n$ and $K_s$ values are higher than typically assigned to open fractures as the vein–wallrock contact is intact.

In these 2D FEM models, there are 24 unknown geomechanical variables that define the wallrock material, vein material, and vein–wallrock contact. The number of unknown parameters (micro-parameters) exceeds the number of known parameters (target macro-properties equivalent to laboratory test results); therefore, the solution is indeterminate [30]. Different combinations of calibrated input values may lead to the same model results, so the calibrated inputs may be one of multiple possible solutions. To reduce the complexity of the required solution, the following assumptions were made.

1.  For all materials, there is zero residual cohesion ($c_r$) or residual tensile strength ($\sigma_{tr}$) after yield as yielding in RS2 represents shear or tensile failure of the material at a mesh node. Although RS2 cannot simulate true detachment of nodes because a FEM model is fundamentally a continuum [26], it can be assumed that these values drop to zero after yield. Based on guidance from Li and Bahrani [30], values of 0.1 MPa were assigned to $c_r$ and $\sigma_{tr}$ in all three materials to avoid numerical convergence errors.

2.  Residual friction angle ($\phi_r$) is equal to the calibrated peak friction angle ($\phi_p$), following guidance from Markus [29] and confirmed by a sensitivity analysis in this study comparing peak strength results, which showed that changing $\phi_r$ by ±10° resulted in a maximum difference in peak strength of just 3% when $\phi_r > \phi_p$ and a 0% difference when $\phi_p > \phi_r$.

3.  For the vein–wallrock contact joint element, the $K_n$:$K_s$ ratio was maintained at 1:1 because the contact is intact [34].

All required input parameters and the final calibrated input properties determined by the methodology explained in this section are listed in Table 3.

### 5.2. Published Test Data of CMET Unit

The laboratory UCS and TCS results utilized for calibration in this study are published by de los Santos Valderrama [23] on mafic rocks without and with sulfide veins from El Teniente mine. The CMET unit is composed of intrusive, hypabyssal rocks of basic composition including gabbros, basaltic porphyry, diabase, and aphanitic rocks [23]. A database of peak strength data from UCS and TCS laboratory tests conducted during 2000–2008 on unveined CMET rock specimens (compiled by [23]) was used to calculate peak strength of unveined CMET rocks for use in this study as a UCS model calibration target. The unconfined peak strength of unveined CMET rocks calculated from this database is 152 MPa, and the peak friction angle and peak cohesion calculated from the UCS and TCS laboratory database of unveined CMET rocks are 32° and 43 MPa, respectively.

**Table 3.** Calibrated input properties for veined UCS and TCS numerical models.

| Base Parameter (Units) | CMET Wallrock Material (w) | Pyrite Vein Material (v) | Vein–Wallrock Contact Joint Element (j) |
|---|---|---|---|
| Unit weight (MN/m$^3$) | $\gamma_{(w)}$ = 0.027 | $\gamma_{(v)}$ = 0.047 | - |
| Young's modulus (GPa) | $E_{(w)}$ = 58.5 | $E_{(v)}$ = 125 | - |
| Poisson's ratio | $\nu_{(w)}$ = 0.14 | $\nu_{(v)}$ = 0.17 | - |
| Peak friction angle (°) | $\phi_{p(w)}$ = 32 | $\phi_{p(v)}$ = 48 | $\phi_{p(j)}$ = 42 |
| Residual friction angle (°) | $\phi_{r(w)}$ = 32 | $\phi_{r(v)}$ = 48 | $\phi_{r(j)}$ = 42 |
| Peak cohesion (MPa) | $c_{p(w)}$ = 43 | $c_{p(v)}$ = 26 | $c_{p(j)}$ = 22 |
| Residual cohesion (MPa) | $c_{r(w)}$ = 0.1 | $c_{r(v)}$ = 0.1 | $c_{r(j)}$ = 0.1 |
| Peak tensile strength (MPa) | $\sigma_{tp(w)}$ = 14 | $\sigma_{tp(v)}$ = 7.5 | $\sigma_{tp(j)}$ = 5 |
| Residual tensile strength (MPa) | $\sigma_{tr(w)}$ = 0.1 | $\sigma_{tr(v)}$ = 0.1 | $\sigma_{tr(j)}$ = 0.1 |
| Normal stiffness (GPa/m) | - | - | $K_{n(j)}$ = 1750 |
| Shear stiffness (GPa/m) | - | - | $K_{s(j)}$ = 1750 |

A subset of published test data was selected for model calibration of a veined CMET specimen based on specimen lithology, vein mineralogy, vein thickness, and vein orientation. All laboratory data used in this study have similar geometries, so a simplified model could be created that accurately reflects key aspects of the physical specimens.

The vein mineralogy in CMET rocks is primarily pyrite with minor chalcopyrite and quartz. The three-veined laboratory specimens selected to calibrate veined UCS and TCS models in this study are VDA-4, VDA-6, and VDA-9, which are shown in Figure 5 with vein geometry, modal mineralogy, and UCS and TCS laboratory test results reported in Table 4. The average vein orientation ($\alpha$) and vein thickness are 26° and 4 mm, respectively, which are used to define the calibration model geometry.

**Table 4.** Characteristics of veined UCS and TCS specimens adapted from [23] and used for model calibration in this study.

| Parameter (Units) | VDA-04 | VDA-06 | VDA-09 |
|---|---|---|---|
| Vein primary mineral % | Pyrite 45% | Chalcopyrite 67% | Pyrite 60% |
| Vein secondary mineral % | Chalcopyrite 27% | Quartz 22% | Chalcopyrite 24% |
| Vein tertiary mineral % | Quartz 24% | Pyrite 6% | Quartz 8% |
| Vein orientation, $\alpha$ (°) | 22 | 20 | 30 |
| Vein thickness (mm) | 4 | 1.5 | 6 |
| Wallrock lithology | CMET | CMET | CMET |
| $\sigma_3$ (MPa) | 5 | 0 | 15 |
| $\sigma_{1peak}$ (MPa) | 91 | 85 | 131 |
| Failure type | Through vein | Through vein | Through vein |
| Angle of rupture relative to core axis (°) | 19 | 22 | 28 |
| $\tau_{peak}$ (MPa) | 29.98 | 27.22 | 50.13 |
| $\sigma_{n(peak)}$ (MPa) | 17.11 | 9.91 | 43.94 |

*5.3. Numerical Sensitivity Analysis*

The objective of this sensitivity analysis is to systematically vary each parameter to investigate their influence on model output. First, the baseline numerical inputs are defined. Each parameter is then incrementally varied, one at a time. The increment is selected to contain a reasonable range of possible input values. Model outputs are recorded and evaluated against the published peak strength laboratory data.

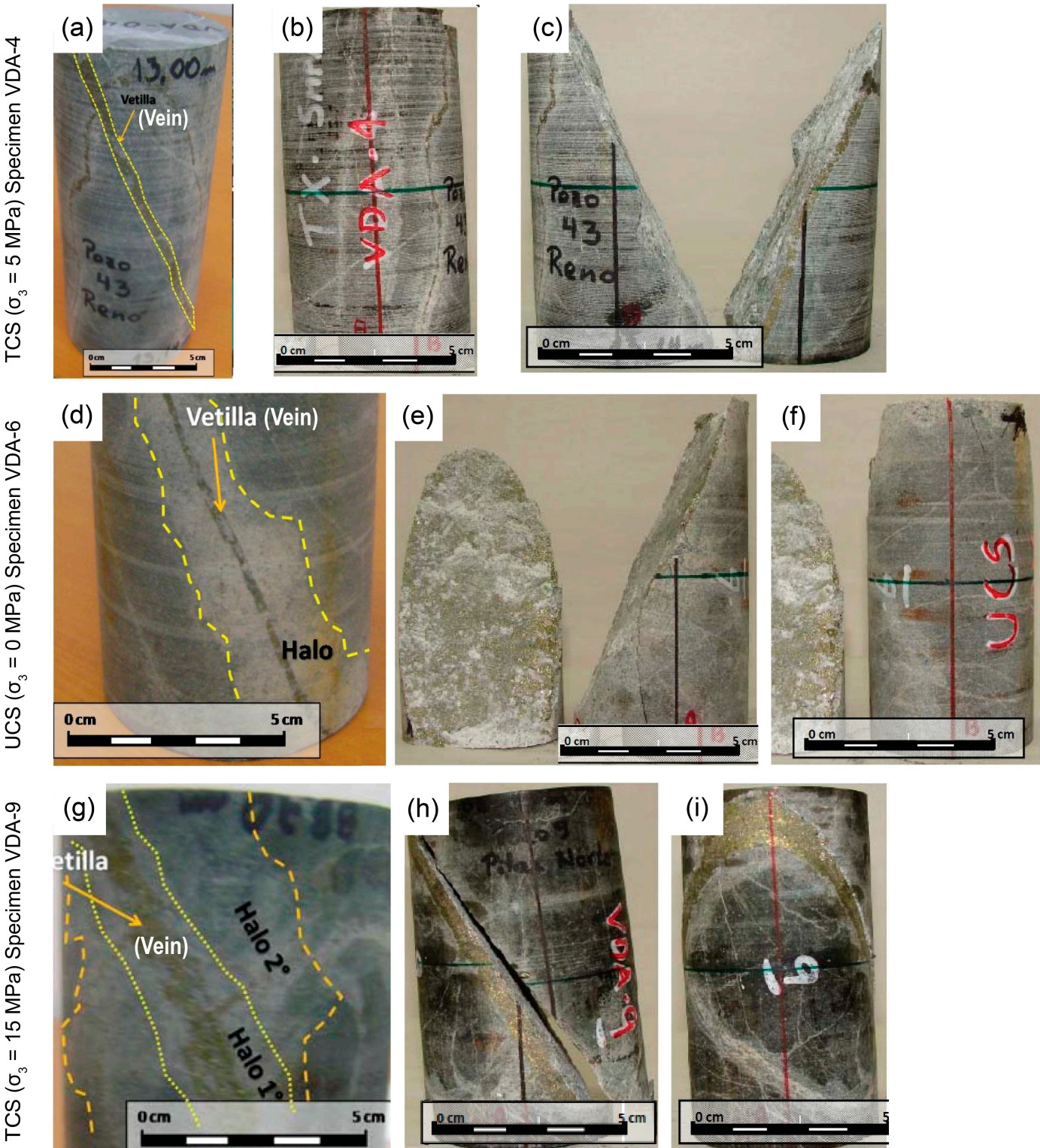

**Figure 5.** Veined UCS and TCS test specimens from the CMET lithological unit: VDA-4 tested at $\sigma_3$ = 5 MPa (**a**) pre-test, (**b**) post-test, (**c**) post-test; VDA-6 tested at $\sigma_3$ = 0 MPa (**d**) pre-test, (**e**) post-test, (**f**) post-test; and VDA-9 tested at $\sigma_3$ = 15 MPa (**g**) pre-test, (**h**) post-test, and (**i**) post-test (modified after [23], photographs reproduced with permission from C.G. de los Santos Valderrama).

Three iterations of sensitivity analyses are used in this study to calibrate the veined UCS test models. The first iteration focuses on the CMET wallrock and pyrite vein material properties. The second iteration focuses on the wallrock–vein contact joint element properties and the vein material properties. Lastly, the third iteration focuses on the wallrock–vein contact joint element and vein material properties in more detail.

### 5.3.1. Selection of Base Case Inputs

Base case values for the vein input parameters, listed in Table 5, were selected from various literature sources for the micromechanical properties of pyrite or an assumed value. Input properties for the vein–wallrock contact joint element are not readily available in the literature. In the case of the veined CMET specimen data utilized in this research, failure during laboratory tests occurred mainly at the vein–wallrock contact. Therefore, Mohr–Coulomb shear strength properties calculated from the test results were implemented as base case inputs for this study.

**Table 5.** Base case model input properties for vein and vein–wallrock contact.

| Model Component | Input Parameter | Assumed Values | Value from Literature | Data from |
|---|---|---|---|---|
| Vein | $E_{(v)}$ (GPa) | | 240 | [36]: 231.1 {14}; [37]: 235 {10}; [38,39]: 306.5 {2}; [40]: 262.8 {1} (Format: average value {# of specimens}) |
| | $\nu_{(v)}$ | | 0.17 | [38,41] |
| | $\phi_{p(v)}$ (°) | | 47 | [41] |
| | $\phi_{r(v)}$ (°) | Equal to $\phi_{p(v)}$ | | |
| | $c_{p(v)}$ (MPa) | | 4.7 | [41] |
| | $c_{r(v)}$ (MPa) | 0.1 | | |
| | $\sigma_{tp(v)}$ (MPa) | | 2 | [41] |
| | $\sigma_{tr(v)}$ (MPa) | 0.1 | | |
| Vein–wallrock contact joint element | $K_{n(j)}$ (MPa/m) | Equal to $K_{s(j)}$ | | Following guidance from [34,35] |
| | $K_{s(j)}$ (MPa/m) | Equal to $K_{n(j)}$ | | Following guidance from [34,35] |
| | $\phi_{p(j)}$ (°) | | 37 | Interpreted from [23] |
| | $\phi_{r(j)}$ (°) | Equal to $\phi_{p(j)}$ | | |
| | $c_{p(j)}$ (MPa) | | 18 | Interpreted from [23] |
| | $c_{r(j)}$ (MPa) | 0.1 | | |
| | $\sigma_{tp(j)}$ (MPa) | | 5 | Interpreted from [23] |
| | $\sigma_{tr(j)}$ (MPa) | 0.1 | | |

### 5.3.2. Sensitivity Analysis: First Iteration

The first iteration of the sensitivity analysis focuses on the CMET wallrock and pyrite vein material properties. In this suite of UCS models, the vein–wallrock contact is modelled using a material boundary which does not require any input parameters. The values of $E$, $\nu$, $\phi_p$, $c_p$, and $\sigma_{tp}$ for the vein material were varied to investigate their influence on peak strength of the whole specimen. The base case input properties were kept constant to isolate one variable which was incrementally changed through a series of models, as shown in Table 6; $\phi_r$, $c_r$, and $\sigma_{tr}$ were assigned the assumed values in Table 5.

The same process was undertaken for all iterations of the sensitivity analysis. For the first iteration, the base case model was created with the values shown in Figure 6a,b. Next, four alternate values for each parameter were selected: minimum, intermediate low, intermediate high, and maximum. A model was created for each value in each parameter where that value was the only one varied. For example, in the minimum value for vein Young's modulus ($E_{(v)}$) case, the numerical inputs are: $E_{(v)}$ = 100,000 MPa (minimum value), $\nu_{(v)}$ = 0.17 (base case), $\phi_{p(v)}$ = 47° (base case), $c_{p(v)}$ = 10 MPa (base case), and $\sigma_{tp(v)}$ = 5 MPa (base case). A total of 40 models were developed for the first iteration of the sensitivity analysis.

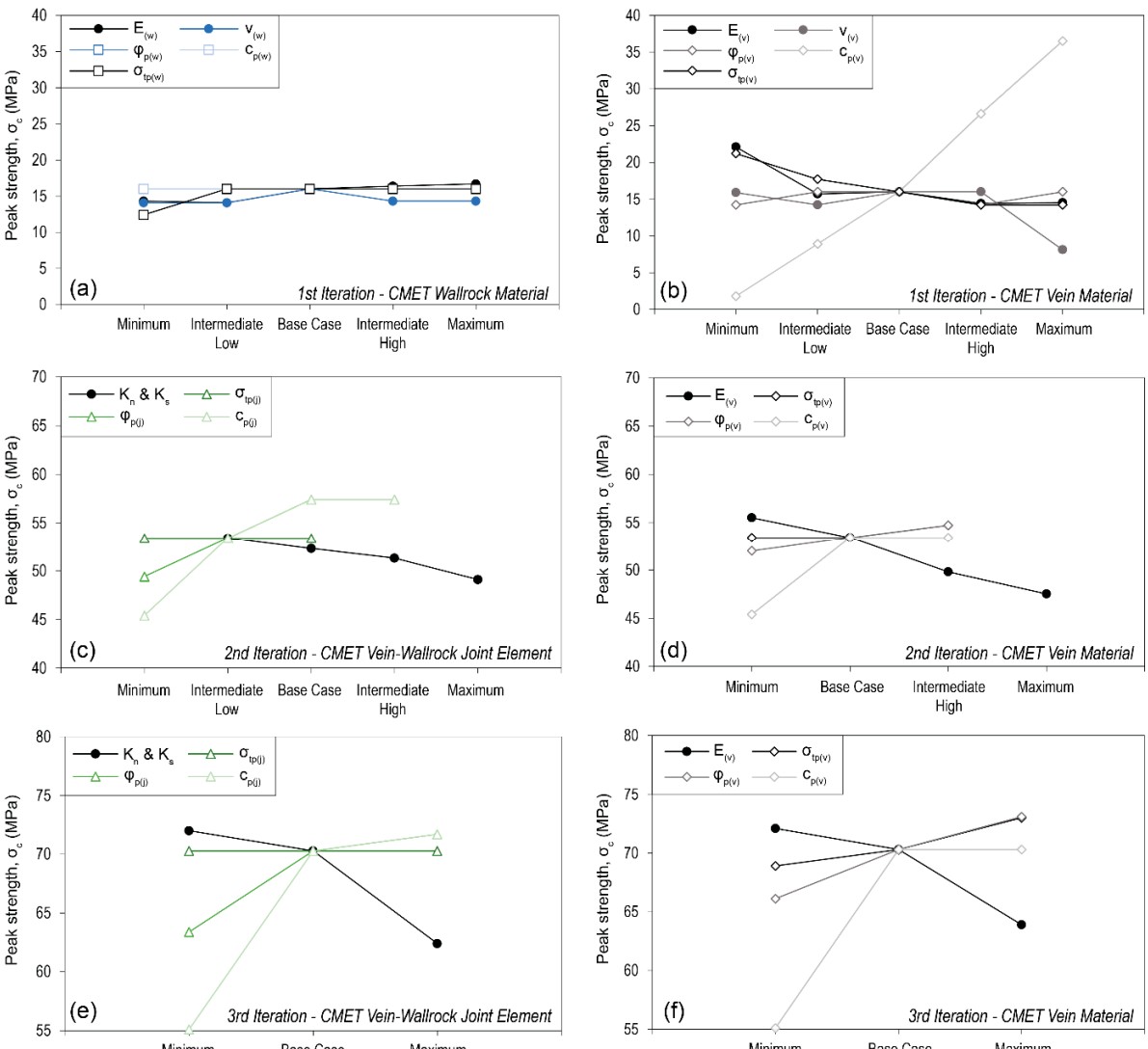

**Figure 6.** Results of sensitivity analyses on CMET UCS models: (**a**) first iteration wallrock material, (**b**) first iteration vein material, (**c**) second iteration vein–wallrock contact joint element, (**d**) second iteration vein material with vein–wallrock contact joint element in use, (**e**) third iteration vein–wallrock contact joint element, and (**f**) third iteration vein material with vein–wallrock contact joint element in use.

The results of this first iteration show that $c_{p(w)}$ and $\phi_{p(w)}$ have no influence on the specimen's peak strength. Increasing $E_{(w)}$ caused a slight increase in peak strength because the ratio of $E_{(v)}$:$E_{(w)}$ was reduced, and low values of $\sigma_{tp(w)}$ caused a decrease in peak strength. Overall, these results indicate that the calibrated wallrock properties from the homogeneous CMET model calibration are suitable because their values have

little influence on the peak strength in UCS models of heterogeneous (veined) CMET rock specimens.

**Table 6.** Numerical inputs for first iteration of sensitivity analysis.

| Parameter | | Minimum | Intermediate Low | 1st Iteration Base Case | Intermediate High | Maximum |
|---|---|---|---|---|---|---|
| Vein material | $E_{(v)}$ (GPa) | 100 | 150 | 205 | 240 | 275 |
| | $\nu_{(v)}$ | 0.13 | 0.15 | 0.17 | 0.19 | 0.21 |
| | $\phi_{P(v)}$ (°) | 37 | 42 | 47 | 52 | 57 |
| | $c_{p(v)}$ (MPa) | 1 | 5 | 10 | 15 | 20 |
| | $\sigma_{tp(v)}$ (MPa) | 1 | 2.5 | 5 | 7.5 | 10 |
| Wallrock (CMET) | $E_{(w)}$ (GPa) | 40 | 50 | 58.5 | 70 | 85 |
| | $\nu_{(w)}$ | 0.1 | 0.12 | 0.14 | 0.16 | 0.18 |
| | $\phi_{P(w)}$ (°) | 22 | 27 | 32 | 37 | 42 |
| | $c_{p(w)}$ (MPa) | 23 | 33 | 43 | 50 | 60 |
| | $\sigma_{tp(w)}$ (MPa) | 8 | 11 | 14 | 17 | 21 |

Based on this set of numerical inputs, the base case model output of peak strength at 16 MPa is significantly lower than the target output (81 MPa). Therefore, values were modified based on the trends exhibiting increased peak strength. As shown in Figure 6b, vein material cohesion, $c_{(v)}$, most significantly influenced peak strength, displaying increases of up to 40% for each 5 MPa increment. Vein friction angle, $\phi_{p(v)}$, exhibited little influence, while decreasing $E_{(v)}$ and $\sigma_{tp(v)}$ caused slight increases in peak strength.

### 5.3.3. Sensitivity Analysis: Second Iteration

For the second iteration of the sensitivity analysis, the base case parameters were re-established, and three additional value sets were tested: minimum, intermediate, and maximum (Table 7). There are significant departures from the literature review values in this effort to calibrate the model to specimen peak strength from physical laboratory UCS test data, particularly $E_v$ and $c_{p(v)}$. It is clear from Figure 6a,b that the base case properties created specimen peak strength far lower than the target peak strength. By decreasing $E_v$, the $E_v:E_w$ ratio decreases and peak strength increases.

**Table 7.** Numerical inputs for second iteration of sensitivity analysis.

| Parameter | | 1st Iteration Base Case | 2nd Iteration Base Case | Minimum | Intermediate | Maximum |
|---|---|---|---|---|---|---|
| Vein–wallrock contact joint element | $K_n$ (GPa/m) | - | 1500 | 2000 | 2500 | 5000 |
| | $K_s$ (GPa/m) | - | 1500 | 2000 | 2500 | 5000 |
| | $\sigma_{tp(j)}$ (MPa) | - | 2.5 | 1 | 5 | - |
| | $\phi_{P(j)}$ (°) | - | 37 | 32 | 42 | 47 |
| | $c_{p(j)}$ (MPa) | - | 18 | 15 | 21 | 24 |
| Vein material | $E_{(v)}$ (GPa) | 205 | 125 | 100 | 150 | 175 |
| | $\nu_{(v)}$ | 0.17 | 0.17 | - | - | - |
| | $\phi_{P(v)}$ (°) | 47 | 2.5 | 1 | 5 | - |
| | $c_{p(v)}$ (MPa) | 10 | 47 | 42 | 52 | - |
| | $\sigma_{tp(v)}$ (MPa) | 5 | 20 | 15 | 25 | - |
| Wallrock (CMET) | $E_{(w)}$ (GPa) | 58.5 | 58.5 | - | - | - |
| | $\nu_{(w)}$ | 0.14 | 0.14 | - | - | - |
| | $\phi_{P(w)}$ (°) | 32 | 32 | - | - | - |
| | $c_{p(w)}$ (MPa) | 43 | 43 | - | - | - |
| | $\sigma_{tp(w)}$ (MPa) | 14 | 14 | - | - | - |

The second iteration base case peak strength is approximately 50 MPa, which is significantly closer to the target (Figure 6c,d). The model components tested in the second iteration are the vein material and vein–wallrock contact modelled as a joint element. The wallrock parameters are not involved in the second iteration as the suitability of the wallrock properties was confirmed by the first iteration sensitivity analysis.

Variation in the vein–wallrock contact joint element properties that caused an increase in specimen peak strength include increased $\phi_{p(j)}$ and $c_{p(j)}$, and decreased $K_n$ and $K_s$. Increasing the vein material $\phi_{p(v)}$ and $\sigma_{tp(v)}$, and decreasing $E_{(v)}$, produced slight increases in peak strength (Figure 6c,d). By decreasing $E_{(v)}$, the $E_{(v)}$:$E_{(w)}$ ratio decreases, which is favourable for increasing specimen peak strength.

5.3.4. Sensitivity Analysis: Third Iteration

The third iteration of the sensitivity analysis began with updated base case properties to further improve the agreement with the target specimen peak strength. Like the second iteration, the third iteration focuses on varying the vein material and vein–wallrock contact joint element properties while maintaining a constant set of wallrock material properties. The third iteration inputs are listed in Table 8. The third iteration base case inputs produce specimen peak strength results that are adequately aligned with the target calibration data (Figure 6e,f). In general, the following trends are observed for the vein material and vein–wallrock joint element parameters.

**Table 8.** Numerical inputs for third iteration of sensitivity analysis.

| Parameter | | Minimum | 3rd Iteration Base Case | Maximum |
|---|---|---|---|---|
| Vein–wallrock contact joint element | $K_n$ (GPa/m) | 1500 | 1750 | 2000 |
| | $K_s$ (GPa/m) | 1500 | 1750 | 2000 |
| | $\sigma_{tp(j)}$ (MPa) | 2.5 | 5 | 7.5 |
| | $\phi_{p(j)}$ (°) | 37 | 42 | 47 |
| | $c_{p(j)}$ (MPa) | 17 | 22 | 27 |
| Vein material | $E_{(v)}$ (GPa) | 100 | 125 | 150 |
| | $\nu_{(v)}$ | - | 0.17 | - |
| | $\phi_{p(v)}$ (°) | 5 | 7.5 | 10 |
| | $c_{p(v)}$ (MPa) | 43 | 48 | 53 |
| | $\sigma_{tp(v)}$ (MPa) | 21 | 26 | 31 |
| Wallrock (CMET) | $E_{(w)}$ (GPa) | - | 58.5 | - |
| | $\nu_{(w)}$ | - | 0.14 | - |
| | $\phi_{p(w)}$ (°) | - | 32 | - |
| | $c_{p(w)}$ (MPa) | - | 43 | - |
| | $\sigma_{tp(w)}$ (MPa) | - | 14 | - |

(i)   The vein–wallrock joint elements control specimen failure when using the base case inputs. This is evidenced when $c_{p(v)}$ is increased, as the peak strength of the specimen does not change. This indicates that it has exceeded the strength of the joint element and therefore increases in material strength cannot improve the peak strength output. Additionally, when $c_{p(j)}$ increases, the peak strength slightly increases. This suggests that while the strength of the joint element may have the most influence on the specimen failure, it is not so significant to cause major increases in peak strength given the critical orientation of the vein in these models.

(ii)  $K_{n(j)}$, $K_{s(j)}$, and $E_{(v)}$ influence the specimen peak strength. When stiffness is increased, the rate at which stress accumulates in the specimen increases and the allowable deformation in the specimen before yield decreases. However, it is important that system stiffness and particularly $K_n$ and $K_s$ are high enough to enable adequate stress transfer through the vein and into the wallrock material below the vein to achieve a realistic numerical simulation.

*5.4. Calibration Results*

The calibrated solution for numerical input properties for the CMET wallrock material, vein material, and vein–wallrock contact joint element in a UCS test simulation based on the three-step sensitivity analysis are listed in Table 3. Incrementally increasing the complexity of numerical features through the three-part iterative sensitivity analysis methodology was necessary to achieve the model output calibration targets.

It is worth highlighting the calibrated solution for $E_{(v)}$ of 125 GPa is significantly lower than the values obtained from the literature (240 GPa, as per Table 5). The literature data are based on crystallographic measurement techniques used to analyze pyrite crystals [36–38,40]. However, the vein material in the model represents a heterogeneous vein material primarily composed of pyrite but also contains other minerals including chalcopyrite and quartz (Table 4), and minor occurrences of anhydrite, based on the physical specimen descriptions from de los Santos Valderrama [23]. This mineral and corresponding stiffness heterogeneity explains the discrepancy in the calibrated solution for $E_{(v)}$.

5.4.1. Validation of Unveined CMET Models

The calibrated solution of input properties was validated in two steps against the physical laboratory test data reported by de los Santos Valderrama [23]. First, the calibrated inputs to simulate homogeneous CMET wallrock (no vein) in UCS test conditions were applied to TCS simulations. Second, the calibrated inputs for CMET wallrock with a pyrite vein from UCS test simulations were applied to TCS test simulations.

A suite of TCS test simulations with confining stresses ($\sigma_3$) of 0, 5, 10, and 15 MPa were conducted on the unveined CMET material. The FEM model results showing major principal stress ($\sigma_1$) contours and yielded material elements in the stage immediately following peak strength are shown in Figure 7a. The peak strength results show close agreement to the El Teniente database of physical laboratory TCS test data on unveined CMET rock specimens, as shown in Figure 7b, where a Mohr–Coulomb failure envelope in principal stress space (Equation (5); [9]) is the linear best fit to the El Teniente database data.

$$\sigma_1 = \left( \frac{2c \cos \phi}{1 - \sin \phi} \right) + \sigma_3 \left( \frac{1 + \sin \phi}{1 - \sin \phi} \right) \tag{5}$$

5.4.2. Validation of Veined CMET Models

The calibrated input properties for the vein material and vein–wallrock interface were used to model the veined specimen in UCS and TCS models at confining stresses ($\sigma_3$) of 0, 5, 10, and 15 MPa. At all four confining stresses, yield in the models occurred as both shear and tensile yield in the vein material and through yielding of the vein–wallrock contact joint element (Figure 8a). The peak strength model results show good agreement with the veined UCS and TCS laboratory test data reported by de los Santos Valderrama [23], specifically specimens VAD-06, VAD-04, and VAD-09, as shown in Figure 8b. The calibrated models and physical specimens VAD-06, VAD-04, and VAD-09 represent the lower bound strength of all laboratory test data from de los Santos Valderrama [23] on veined CMET specimens, which is attributed to the critically inclined orientation of the single vein in each specimen. Failures in both the physical tests and numerical models occurred through the vein material and vein–wallrock contact, as shown in Figures 5 and 8, respectively.

As a result of this validation against UCS and TCS laboratory test data, the UCS and TCS models with calibrated input properties (Table 3, Figure 8) provide a realistic basis to conduct the numerical TCS experiments on veined specimens discussed in the following sections, which investigates the influence of vein thickness and orientation on the resulting specimen strength and yield behaviour.

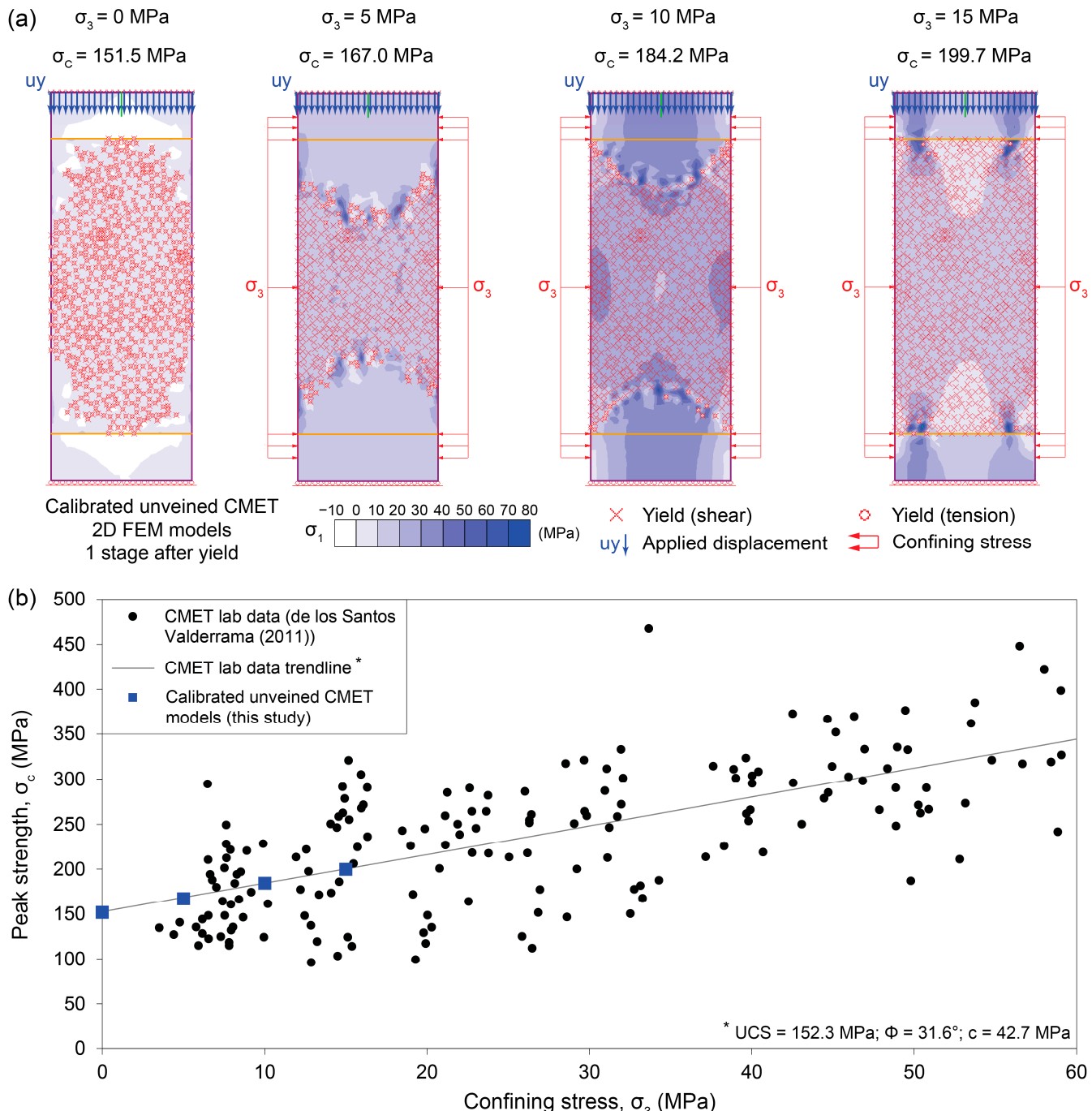

**Figure 7.** (**a**) Calibrated unveined UCS and TCS model results of CMET rock from this study in stage immediately following peak strength ($\sigma_c$); (**b**) El Teniente mine physical laboratory test database with linear failure envelope and reported Mohr–Coulomb properties (data from de los Santos Valderrama [23]), and compared to calibrated unveined CMET numerical models.

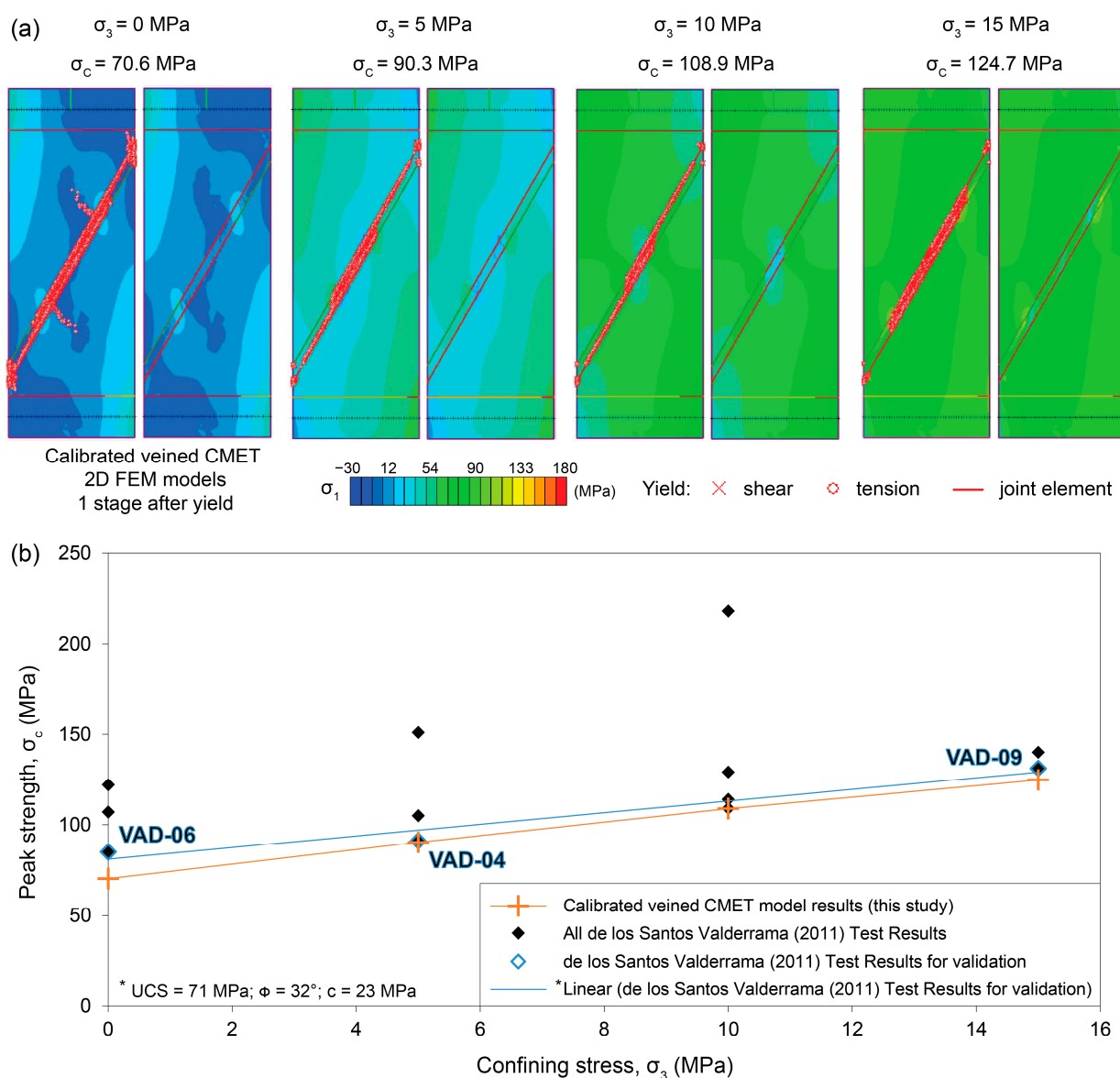

**Figure 8.** (**a**) Calibrated veined CMET UCS and TCS model results in stage immediately following peak strength ($\sigma_c$); (**b**) compared to physical laboratory test data from de los Santos Valderrama [23] with linear failure envelope and reported Mohr–Coulomb properties.

## 6. Numerical TCS Experiments on Veined CMET

The numerical UCS and TCS investigation in this study aims to determine how the geometric features of veins in rock specimens influence the measured geomechanical properties. Rock specimens with single, isolated veins are the focus, where both the vein thickness and vein orientation are systematically varied. This detailed array of vein geometry is not typically possible to achieve in a physical laboratory testing program where the range of representative specimens selected for testing is limited by the varieties of geological characteristics exhibited in samples available from geotechnical diamond drilling programs [7]. Even in cases where thorough drilling and sampling programs have been conducted, it is often impractical to collect and test specimens representing the full range of geometries due to the nature of sampling and budget constraints.

### 6.1. Modelling Program

The calibrated and validated numerical model configuration of a veined UCS specimen presented in the previous sections is utilized as the baseline of the models created for the

following experiment, where major components of the models include CMET wallrock material, sulfide vein material, two vein–wallrock contact joint elements, top and bottom steel platens, and two platen–specimen contact joint elements. A summary of this numerical UCS and TCS experimental program is provided in Table 9.

**Table 9.** Summary of numerical UCS and TCS experiment program.

| Confining Stress, $\sigma_3$ (MPa) | Vein Thickness (mm) | Vein Orientation, $\alpha$ (°) | Increments of Vein Orientation between Models (°) |
|:---:|:---:|:---:|:---:|
| 0 | 4 | 10–80 | 10 |
| 5 | 4 | 10–80 | 10 |
| 10 | 1, 4, and 8 | 5–85 | 5 |
| 15 | 4 | 10–80 | 10 |

*6.2. Numerical Experiment Results*

The results of the numerical UCS and TCS experiments at $\sigma_3$ confinements of 0, 5, 10, and 15 MPa are evaluated based on the elastic response, failure type, and peak strength of the specimens.

6.2.1. Specimen Elastic Response

A decrease in Young's modulus of the veined specimen with respect to increasing vein orientation ($\alpha$) was observed in this numerical experiment. Although the elastic behaviour of the vein and wallrock materials are each consistent and in good agreement with their respective input properties, the elastic behaviour of the whole veined specimen is influenced by the vein–wallrock contact joint element, as illustrated in Figure 9. When the vein is oriented between 0° and 20°, the vein ends at the top and bottom of the specimen which is in contact with the platens. In this scenario, the stiffness of the overall specimen is controlled by the stiffer wallrock which is in contact with both platens across the entire specimen and uninterrupted by the vein. When the vein orientation is between 25° and 90°, the vein ends on the specimen sides and is no longer in contact with the platens. As a result, the applied load is transferred through both the wallrock and vein materials across the height of the specimen. In this scenario, the emergent specimen stiffness is a combination of behaviours from both the stiffer wallrock and softer vein materials, as well as the joint element normal stiffness and shear stiffness of the vein–wallrock contacts.

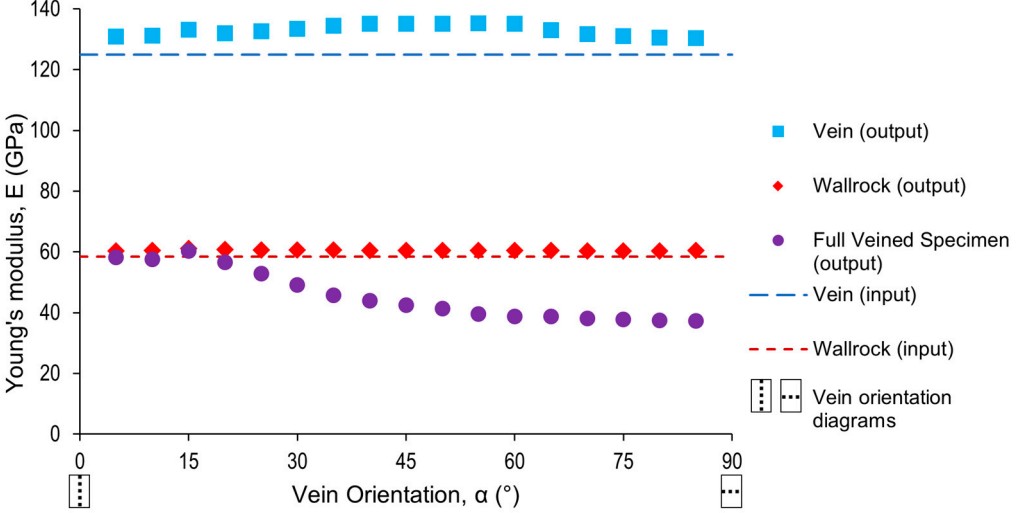

**Figure 9.** Elastic response of veined TCS models at 10 MPa of confining stress ($\sigma_3$) with a vein thickness of 4 mm.

### 6.2.2. Failure Types

Three types of failure were observed in the numerical UCS and TCS experiments, which are classified herein as Type A, Type B, and Type C, with the following definitions. Examples of each failure type are shown in Figure 10.

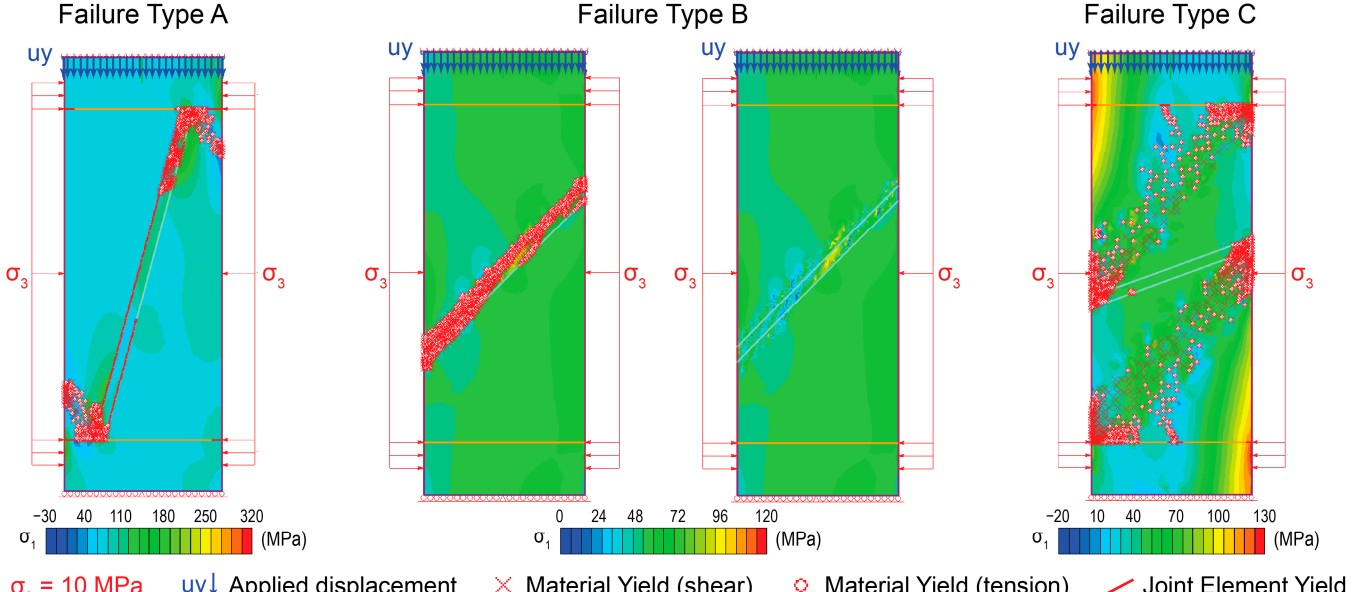

**Figure 10.** Examples of failure Types A, B, and C observed in veined UCS and TCS models.

(i) Type A failures: Failure primarily occurs through both the vein material and vein–wallrock contact joint element. Minor material yield in the wallrock may occur near the ends of the specimen.

(ii) Type B failures: Failure primarily occurs through the vein material but not the vein–wallrock contact joint element. Minor material yield in the wallrock may occur near the vein–wallrock contact.

(iii) Type C failures: Failure occurs in wide bands of predominantly shear through the wallrock material on one or both sides of the vein. Some failure may occur in the vein material, but no failure occurs in the vein–wallrock contact joint element.

The results of this experimental program show Type A failures largely occur at vein $0° < \alpha < 25°$ from the core axis. At $35° < \alpha < 45°$, Type B failure dominates, and at $55° < \alpha < 85°$ (sub-perpendicular to core axis), Type C failure dominates. Models with vein orientations of 25–35° and 45–55° exhibit transitionary failure behaviours between Types A to B and types B to C, respectively. For these ranges of vein orientation, there is little change in failure type between the tested vein thicknesses (1, 4, and 8 mm). These results are illustrated in Figures 11 and 12.

To evaluate the effect of veined specimen failure type on strength in the full suite of UCS and TCS models, the generalized Hoek–Brown strength criterion (Equations (6)–(9); [9]), where $GSI = 100$ and $D = 0$, was utilized to quantify strength envelopes for each failure type (Figure 13).

$$\sigma_1 = \sigma_3 + \sigma_{ci}\left(m_b\frac{\sigma_3}{\sigma_{ci}} + s\right)^a \tag{6}$$

$$m_b = m_i exp\left(\frac{GSI - 100}{28 - 14D}\right) \tag{7}$$

$$s = exp\left(\frac{GSI - 100}{9 - 3D}\right) \tag{8}$$

$$a = \frac{1}{2} + \frac{1}{6}\left(e^{-GSI/15} - e^{-20/3}\right) \tag{9}$$

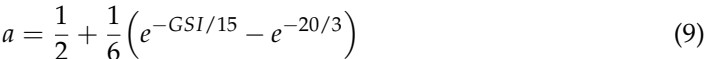

**Figure 11.** Veined TCS models at 10 MPa confining stress ($\sigma_3$), with vein orientations between 5° and 45°, at the stage immediately following peak strength, sorted by vein thickness and orientation; failure types and transitions are indicated.

Specimens that exhibited failure Types A and B produced similar Hoek-Brown strength envelopes that are weaker than that of Type C failures. All three strength envelopes have unconfined compressive peak strength ($\sigma_{ci}$) values of between 74 and 81 MPa. The $m_i$ value for the failure Type C envelope is significantly higher (12.7) than that of the Type A and B envelopes (7.2 and 4.5, respectively).

### 6.2.3. Effects of Vein Orientation and Thickness

The changes in failure type with respect to different vein orientations in the UCS and TCS models contribute to the variation in specimen peak strength with different vein orientations (Figure 14). At vein orientations near parallel to the core axis and specimen loading direction (close to 0°), peak strength is approximately 25% to 38% lower than the unveined CMET specimen (184 MPa). As vein orientation increases, peak strength

decreases to a minimum of approximately 100 MPa, or 45% lower than the unveined CMET specimen, at vein orientations from 15° to 30°. At vein orientations greater than 30°, peak strength increases until maximum peak strength of the veined specimens is achieved at vein orientations of 80° to 85°. In this scenario, however, the veined specimen peak strength is approximately 8% to 36% lower than the unveined CMET specimen. This general effect of vein orientation agrees with Clark and Day's [7] physical laboratory test data on the Legacy skarn black granodiorite unit with weakening carbonate veins, as well as Jaeger and Cook's [42] model for impacts of anisotropic foliation orientation on laboratory UCS and TCS specimen peak strengths.

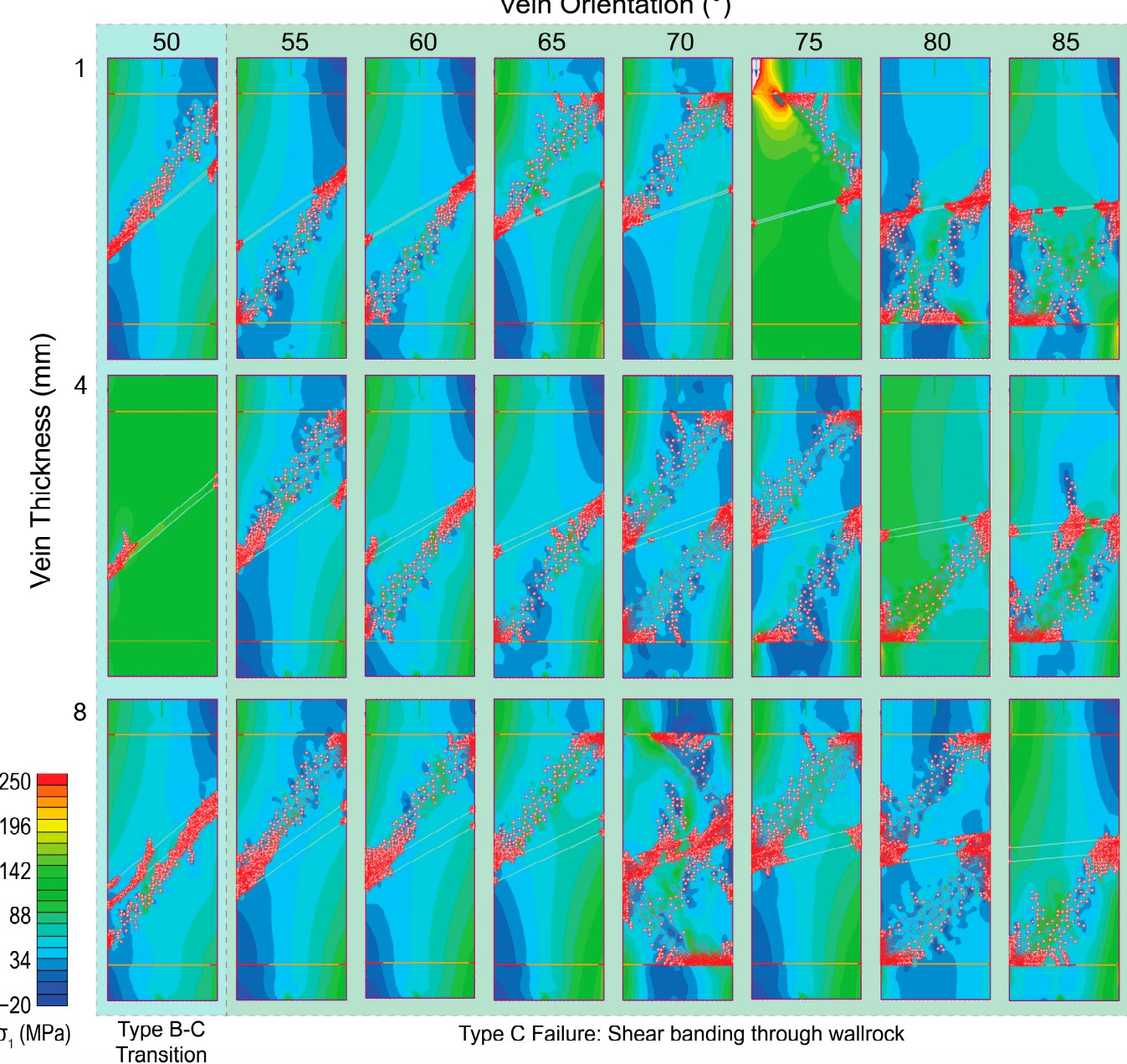

**Figure 12.** Veined TCS models at 10 MPa confining stress ($\sigma_3$), with vein orientations between 50° and 85°, at the stage immediately following peak strength, sorted by vein thickness and orientation; failure types and transitions are indicated.

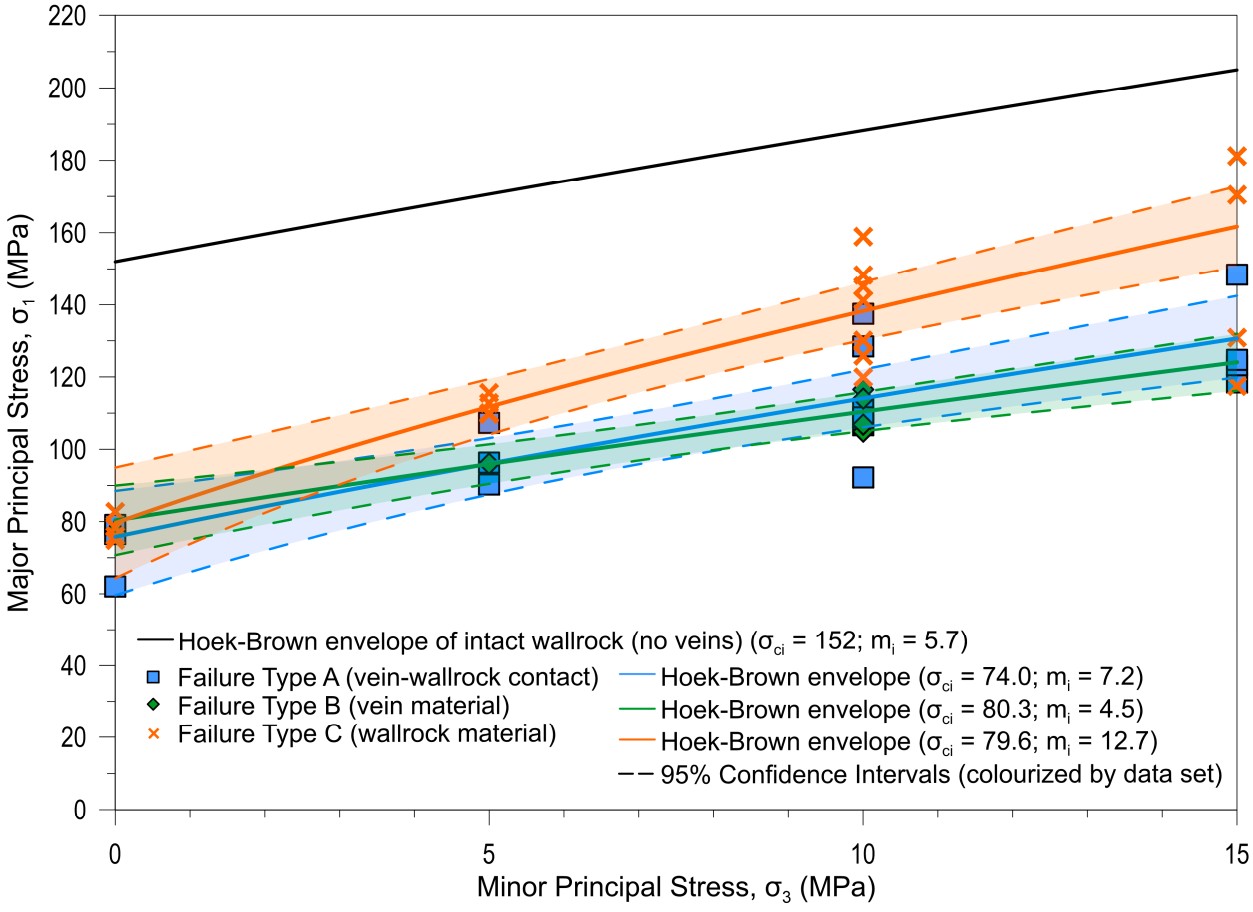

**Figure 13.** Hoek–Brown strength envelopes with 95% confidence intervals for each veined CMET model grouped by specimen failure type and for the unveined CMET specimens.

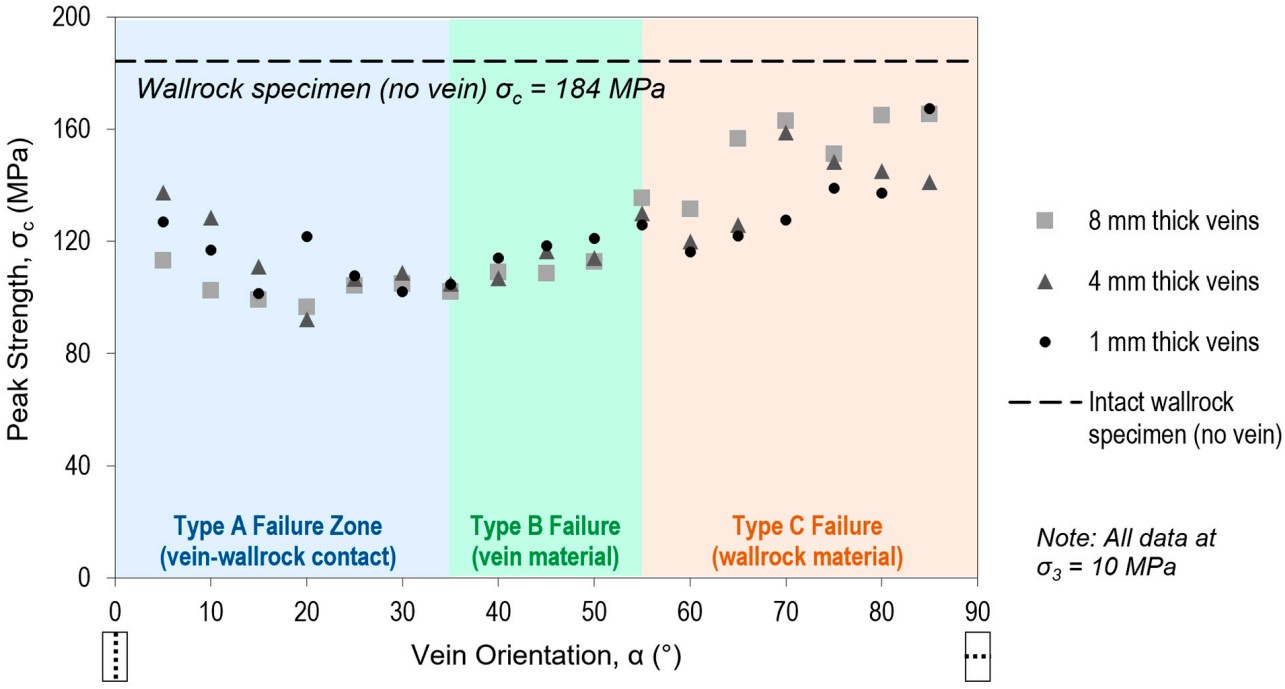

**Figure 14.** Vein orientation and thickness effects on peak strength at $\sigma_3 = 10$ MPa confining stress.

Vein thickness also exhibits a notable influence on specimen peak strength (Figure 14). Generally, specimens with the thickest (8 mm) veins were weakest at vein orientations less than 25° (Type A failures) but strongest at vein orientations greater than 55° (Type C failures), and the opposite occurred for specimens with the thinnest (1 mm) veins. In these two ranges of vein orientations, significant variability in specimen peak strength between vein thickness occurred of up to approximately 40 MPa. At vein orientations from 30° to 50° (Type A and B failures), little influence of vein thickness on specimen peak strength occurred, where the variability in peak strength was up to approximately 15 MPa.

Confining stress exhibits significant influence on peak strength of veined UCS and TCS models. Higher confinement generally increased peak strength across all vein orientations. Furthermore, at vein orientations less than 25° and greater than 65°, high confinement ($\sigma_3$ = 10–15 MPa) produced significantly larger peak strengths compared to vein orientations between 25–65°. In contrast, there is little variability in peak strength with different vein orientations at low confinement ($\sigma_3$ = 0–5 MPa) (Figure 15).

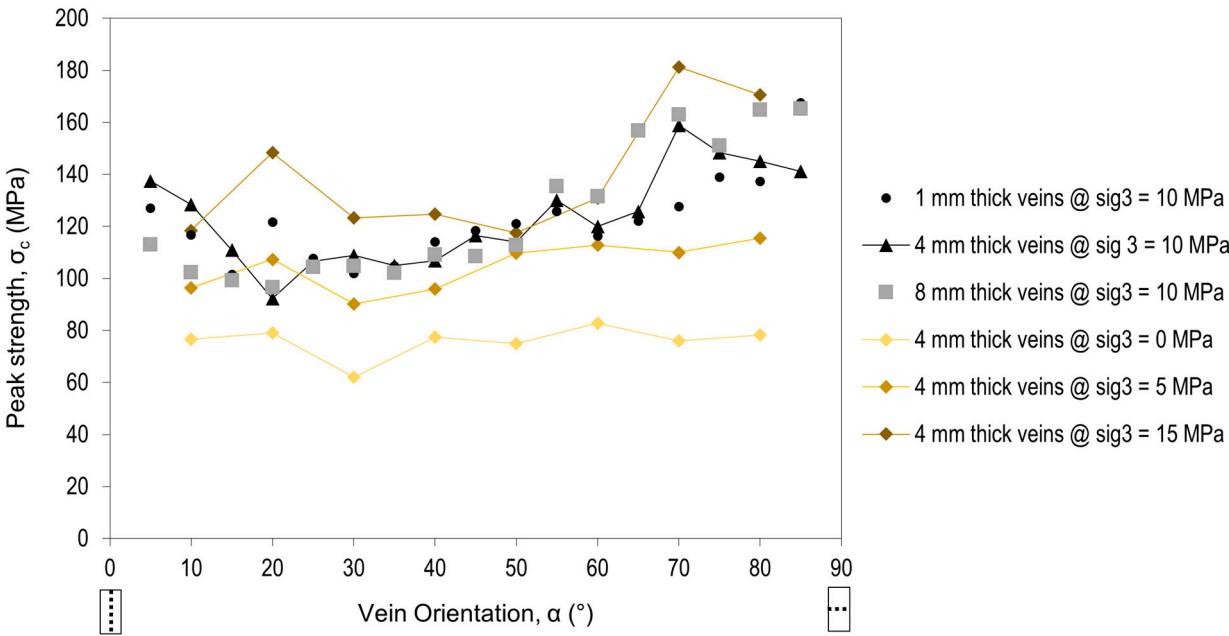

**Figure 15.** Influence of confinement ($\sigma_3$) on peak strength response for 4 mm-thick veins with respect to vein orientation.

### 6.2.4. Intact Vein Shear Strength

Shear strength of veins is an important geomechanical parameter, as it can be used to define numerical inputs for veins that are modelled as individual pseudo-discrete elements in FEM, discrete elements in discrete element method software, or similar geomechanics software. Physical laboratory direct shear testing of intact veins presents various challenges associated with relative strengths of the target vein, wallrock, and encapsulating material (cement grout) [20], so measurement of vein shear strength from TCS tests provides a valuable alternative [19]. In this study, the shear strength of each intact vein was calculated for each specimen where failure occurred through the vein using the Mohr–Coulomb shear strength criterion. To quantify the shear strength of veins, it is essential to use a strength criterion that can accommodate nonzero cohesion and tensile strength. The normal and shear stresses of the failure plane parallel to the failed vein were calculated based on the orientation of the major ($\sigma_1$) and minor ($\sigma_3$) principal stresses imposed by the TCS test loading conditions, using on Equations (10) and (11) [43].

$$\sigma_n = \frac{\sigma_1 + \sigma_3}{2} - \frac{\sigma_1 - \sigma_3}{2} \cos 2\beta \tag{10}$$

$$\tau = \frac{\sigma_1 - \sigma_3}{2} \sin 2\beta \qquad (11)$$

The vein shear strengths for 4 mm-thick veins are plotted with Mohr–Coulomb strength envelopes for each group of data sorted by specimen failure type (Figure 16a), vein orientation (Figure 16b), and confining stresses (Figure 16c).

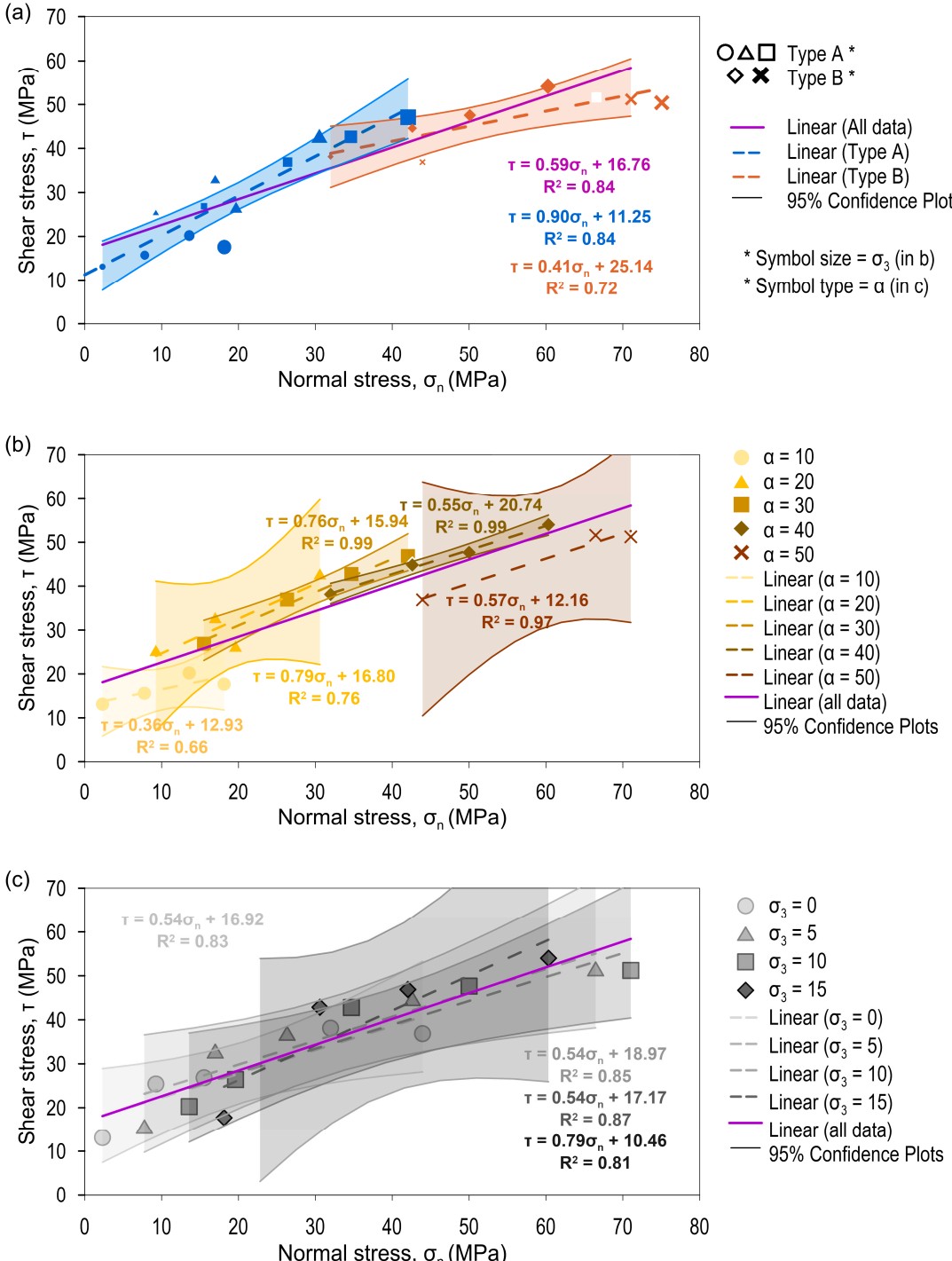

**Figure 16.** Mohr–Coulomb shear strength envelopes of veins calculated from numerical TCS test results for vein thickness of 4 mm, inclined at 10, 20, 30, 40, and 50 degrees, and with confining stresses of 0, 5, 10, and 15 MPa, sorted by (**a**) failure type, (**b**) vein inclination, and (**c**) confining stress, and showing 95% confidence intervals.

The data sorted by failure type (Figure 16a) include only failure Types A and B, which involved failure through the vein material and/or vein–wallrock contact joint element. Type C failure data are excluded from this analysis of vein shear strength, as failure occurred through the wallrock material. The Mohr–Coulomb envelopes sorted by failure type are reminiscent of Patton's [44] bilinear shear strength criterion for discontinuities, with low cohesion and high friction angle at low confinements with a transition to the opposite at higher normal stresses.

Regarding vein orientation (Figure 16b), only veins at 10° to 50° were included in this analysis due to the exclusion of specimen failure Type C data, which did not exhibit vein failure. Specimens with vein orientations of 30° to 40° produced Mohr–Coulomb envelopes that are most representative of the entire data set and have the tightest 95% confidence intervals, indicating a statistical preference for measuring shear strength of veins tested in these orientations. This aligns with the observation that failure in specimens with veins oriented between 30° and 40° occurred through the vein with little to no yield of the wallrock (Figure 11). Shear strength data from veins at 10°, 20°, and 50° orientations represent less than half of the normal stress range included in the full data set and exhibit relatively wide 95% confidence intervals.

In Figure 16c, the data set and Mohr–Coulomb envelope at each confining stress include vein orientations between 10° and 50° at 10° increments. As the applied normal stress component on the vein surface changes with different vein orientations, a relatively large range of normal stress is obtained for a group of specimens tested at the same confining stress. Additionally, the friction angle for the group of specimens at confinements of 0, 5, and 10 MPa is identical (28.4°), and the cohesion varies by only 1 to 2 MPa. The 95% confidence intervals tend to widen with increasing confining stress, suggesting a slight statistical preference toward vein shear strength data collected from lower confining stress test programs.

## 7. Discussion

### 7.1. Model Evaluation

The primary method utilized in this study to validate the UCS and TCS numerical models is the comparison of macroscopic model outputs to physical laboratory test results and analytical solutions. Post-peak behaviour cannot be explicitly represented using continuous FEM due to displacement compatibility requirements restricting internal element displacement (i.e., nodal displacement) such that the body remains continuous before and after deformation [26,45]. Given the nature of the numerical simulations in this research, nonconvergence is expected after material yield, particularly in UCS models where strains larger than infinitesimal scale often occur after yield. The computation of each model stage finishes when the termination criteria is met. Given that an approximate solution is obtained numerically in FEM models, the allowable tolerance for the termination criteria is defined. Nonconvergence occurs when the termination criteria are not met, which is implemented to avoid unreasonably long computation times. The fracturing of an initially intact medium in this work, which results in the release and imbalance of forces, creates a nonconvergent condition that realistically represents the physical phenomenon.

Determining the stage of yield onset in the models in this study is based on the accumulation of yielded mesh and/or joint elements, as well as examination of axial stress-strain data for various monitoring points in the centre of the modelled specimen. It is important to note that each of these yield determinations are not reliable individually because a small number of yielded elements prior to the complete failure of the specimen may occur, particularly in TCS models. Yielding of nodes or segments of joint elements prior to model yielding is interpreted as micro-fracturing prior to reaching the peak strength, and it is the accumulation of these yielded elements that causes yielding of the whole specimen.

### 7.2. Calibration Results

Overall, the trends of parameter sensitivity are similar for both the vein material and the joint element. For the vein material, the most sensitive parameter is Young's modulus. When decreased, the ratio between it and the wallrock Young's modulus decreases and the specimen peak strength increases. Increasing tensile strength, friction angle, and cohesion also result in an increase of specimen peak strength.

The most sensitive parameters in the joint element are cohesion and normal and shear stiffnesses. Increasing cohesion causes the specimen peak strength to increase, and increasing both normal and shear stiffness causes the specimen peak strength to decrease.

### 7.3. Numerical UCS and TCS Experiments

For the numerical experiments on veined CMET specimens, the elastic responses of the vein and wallrock align with the numerical inputs for each material (Figure 9). For the whole specimen Young's modulus, as the orientation of the vein increases (from parallel to perpendicular to core axis and loading direction), the Young's modulus decreases. This is explained by the corresponding decrease in vertical thickness of the vein and the application of major principal stress becomes more perpendicular to the vein, decreasing shear stress and correspondingly increasing normal stress application. In this position, the normal stiffness of the vein has more control on the overall stiffness of the specimen. These results agree with the findings from veined laboratory UCS tests by Clark and Day [7].

The influence of vein thickness changes with respect to vein orientation. The effect of thickness is least significant when veins are oriented from $20° < \alpha < 50°$. As vein orientation increases toward $90°$, thinner veins decrease specimen peak strength more significantly than specimens with thicker veins for specimens that fail through the wallrock. This may be due to closer vein–wallrock contact joint elements in thinner veins and may be a function of the vein in this study being softer and weaker than the wallrock; additional research is warranted to examine this behaviour in laboratory studies where brittle fracture mechanics can be directly observed and for different ratios of vein: wallrock strengths.

Veined models exhibiting Type C failures, where yield occurs as shear bands in the wallrock around veins oriented $55° < \alpha < 90°$, generally exhibit a lower strength than the unveined models (Figure 14). This indicates that the presence of veins that are weaker than the wallrock, even when not critically oriented, reduces the strength of UCS and TCS test specimens. The impact of vein orientation on specimen peak strength in this study partly agrees with the Jaeger and Cook [42] model, as summarized in Table 10. There is good agreement between the range of critical vein orientations where failure occurs along the discontinuity, and when failure occurs along foliation in the Jaeger and Cook [42] model. However, two differences where this study on veins disagrees with Jaeger and Cook's [42] work on foliated rocks are:

(1)  this study shows peak strength of the specimen is reduced even by the presence of non-critically oriented weakening veins ($55° < \alpha < 90°$);

(2)  this study shows the vein orientation resulting in the weakest peak strength ranges from $20$–$30°$, depending on the magnitude of confining stress ($\sigma_3$).

**Table 10.** Comparison between this study and data from [42].

| Comparison Item | Data from Jaeger and Cook [42] | Numerical UCS/TCS Experiment Results from This Study |
|---|---|---|
| Discontinuity type | Foliation | Single hydrothermal vein |
| Range of discontinuity orientation where failure occurs on discontinuity, i.e., critical angles ($\alpha$, °) | 10–52 | 5–55 |
| Weakest orientation ($\alpha$, °) | 30 | 20–30 (for different $\sigma_3$) |
| Influence of discontinuity outside critical angles | No influence | Decreases specimen peak strength |

These findings agree with the laboratory test results by Clark and Day [7]. Thus, it is important to recognize the Jaeger and Cook [42] model for foliated rocks cannot be directly applied to hydrothermally veined rocks. Further investigation on veined rocks with different ratios of wallrock and vein strength is needed to explore the complex relationship between UCS/TCS specimen strength and vein orientation.

If the strength of the vein material were increased such that it was stronger than the wallrock material, then the strength of the UCS/TCS specimens may increase compared to the unveined specimen, as demonstrated by Clark and Day [7]. Primary or secondary alteration halos [46] are additional geological features around the vein–wallrock contact that may be an important control on specimen strength, which requires further research.

The Hoek–Brown strength envelopes shown in Figure 13 represent the whole specimen results, as opposed to the shear strength of the veins. Calculating these failure envelopes on results from this study of unveined specimens as well as single weakening veined specimens provides upper and lower bound Hoek–Brown rockmass strength envelopes, respectively. This is useful for direct input to material properties for excavation scale numerical models where veins are implicitly modelled structures in an equivalent continuum rockmass material.

Three primary failure mechanisms were observed for UCS and TCS specimens with single veins across different orientations: Type A and Type B exhibit failure through the vein and collectively occur at $0° < \alpha < 50°$, and Type C exhibits failure through the wallrock and occurs at $50° < \alpha < 90°$. Sorting vein shear strength results by failure type produces a bi-linear envelope akin to Patton [44] where Type A failures exhibit low cohesion, high friction, and occur at low normal stresses and Type B failures exhibit high cohesion, low friction, and occur at high normal stresses (Figure 16a). When measuring shear strength of a vein from TCS results, testing a suite of specimens with the same vein orientation at different confining stresses produces a shear strength envelope across a small range of normal stresses (Figure 16b). In contrast, testing a suite of TCS specimens with a variety of vein orientations but small range of confining stress produces a shear strength envelope with a significantly larger range of normal stresses (Figure 16c). This provides important insight for planning effective TCS laboratory testing programs where measuring the shear strength of single veins is an objective.

Further research is recommended to investigate the effects of different ratios of wallrock to vein strength on the range of vein orientations and thicknesses considered in this study, particularly when veins are stronger than the wallrock and the wallrock–vein contact is strongly welded. Complexities regarding the vein microstructure and vein mineral growth patterns may also influence the emergent stiffness and strength of veined specimens.

## 8. Conclusions

The calibrated input properties determined in this study through a suite of numerical FEM UCS and TCS models with confining stresses ($\sigma_3$) of 0, 5, 10, and 15 MPa represent a geomechanical solution to model unveined Lac du Bonnet granite as well as veined CMET rocks from El Teniente mine with pyrite $\pm$ quartz $\pm$ chalcopyrite vein mineralization. Incrementally increasing the complexity of numerical features through the iterative calibration methodology was required to achieve the desired calibration of input properties to the target physical laboratory UCS and TCS test data. The calibrated solutions were validated against physical UCS and TCS laboratory test data reported by Labeid [22] and Martin [21] for LdB granite and de los Santos Valderrama [23] for the CMET rocks.

The numerical TCS experiments on CMET rock specimens with single veins in this study demonstrate the presence of veins weaker than the wallrock, even when not critically oriented (i.e., $50° < \alpha < 90°$), soften and weaken the overall specimen. At confining stresses of 10 MPa, veins oriented between $25° < \alpha < 50°$ had the most detrimental impact on specimen peak strength, and failure primarily occurred through the vein material. Vein thickness plays a variable influence on peak strength. For steeply oriented veins ($0° < \alpha < 20°$), thicker

veins (8 mm) exhibit the greatest reduction on specimen peak strength. For veins with intermediate vein orientations, the influence of thickness is small relative to the impact of orientation at these critical angles ($20° < \alpha < 55°$). For veins nearing sub-perpendicular orientation to the core axis and load direction ($55° < \alpha < 90°$), thin veins (1 mm) cause the greatest reduction in specimen peak strength. Vein orientation had the least impact on specimen peak strength at zero confining stress ($\sigma_3 = 0$ MPa).

For geotechnical laboratory testing programs where TCS testing is utilized to determine the shear strength of veins or other intrablock structures, the results of this study highlight the importance of testing a suite of specimens that are representative of the rockmass, including with discrete veins at various orientations. During the laboratory sample selection stage of a site investigation program, it is important to consider the numerical modelling techniques that will be used later in the rock engineering design process. Specifically, if veining occurs as a ubiquitous stockwork network, it may be impossible to select samples without veins to test the strength of the wallrock, or to select veined specimens with single, isolated veins. Therefore, the veined rock should be considered as a primary intact rock type and included in sample selection for laboratory testing.

If core samples with veins can be obtained, and modelling veins as discrete or pseudo-discrete structures is desired [11], the following sample selection and test program is recommended. A minimum of 8 specimens per lithology and vein mineralogy should be tested under TCS conditions. The range of vein orientations should be spread as much as possible between $5° < \alpha < 50°$. These results should be analyzed such that the failure type of each test is considered, and the calculated vein Mohr–Coulomb shear strengths be used to determine the vein failure envelope. If the failure mechanism is consistent (e.g., Type A only), then all test results should be utilized to determine the extent of the failure envelope. If two failure mechanisms are observed, categorization of the failure methods should be used to sort the shear strength data, and alternative nonlinear or bilinear failure envelopes should be considered. Analyzing a suite of veined rock TCS data to calculate Hoek–Brown intact rock strength envelopes, sorted by wallrock lithology, vein mineralogy, and TCS test failure type will alternatively provide inputs to material properties where veins are modelled as part of the continuum material.

**Author Contributions:** Conceptualization, J.J.D.; methodology, G.A.R. and J.J.D.; investigation, G.A.R.; formal analysis, G.A.R. and J.J.D.; funding acquisition, J.J.D. and G.A.R.; software, J.J.D.; supervision, J.J.D.; visualization, G.A.R. and J.J.D.; writing—original draft, G.A.R.; writing—review and editing, J.J.D. All authors have read and agreed to the published version of the manuscript.

**Funding:** This research was funded by the Natural Sciences and Engineering Research Council of Canada (Day Discovery Grant; Rudderham Canadian Graduate Scholarship—Master's), Queen's University, and the University of New Brunswick.

**Data Availability Statement:** The data presented in this study are available from the corresponding author by reasonable request.

**Acknowledgments:** For many technical discussions, thanks to Mark Diederichs, Jean Hutchinson, Chris Spencer, Timothy Packulak, Joseph White, Adrian Park, Steven Hinds, and Émelie Gagnon.

**Conflicts of Interest:** The authors declare no conflict of interest.

## Abbreviations

| | |
|---|---|
| 2D | 2-dimensional |
| 3D | 3-dimensional |
| c | Cohesion (MPa) |
| CMET | Complejo Máfico El Teniente/El Teniente Mafic Complex |
| E | Young's modulus (MPa) |
| $E_{2D}$ | Young's modulus for plane strain analysis (MPa) |

| $E_{3D}$ | Young's modulus measured from physical laboratory test (MPa) |
|---|---|
| FEM | Finite element method |
| ISRM | International Society for Rock Mechanics and Rock Engineering |
| $K_n$ | Joint normal stiffness (MPa/m) |
| $K_s$ | Joint shear stiffness (MPa/m) |
| LdB | Lac du Bonnet granite |
| TCS | Triaxial compressive strength |
| UCS | Uniaxial (or Unconfined) compressive strength |
| $\alpha$ | Angle between core axis and vein in diamond drill core |
| $\gamma$ | Unit weight (MN/m$^3$) |
| $\mu$ | Coefficient of friction |
| $\nu$ | Poisson's ratio |
| $\nu_{2D}$ | Poisson's ratio for plane strain analysis |
| $\nu_{3D}$ | Poisson's ratio measured from physical laboratory test |
| $\phi$ | Friction angle (degrees, °) |
| $\sigma_1$ | Major principal stress (MPa) |
| $\sigma_3$ | Minor principal stress (MPa) |
| $\sigma_c$ | Peak compressive strength (MPa) |
| $\sigma_n$ | Normal stress (MPa) |
| $\tau$ | Shear stress (MPa) |

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
