# Peer review of "Veined Rock Performance under Uniaxial and Triaxial Compression Using Calibrated Finite Element Numerical Models"

_geotechnics, doi:10.3390/geotechnics3040067_

Round 1

Reviewer 1 Report

Comments and Suggestions for Authors

This paper provides a framework to evaluate impacts of any orientation and thickness of a single vein on stiffness and strength of laboratory scale specimens under uniaxial and triaxial compression using finite element numerical experiments. This methodology greatly improves the value of physical laboratory test data with a limited scope of vein characteristics in the tested specimens by using calibrated numerical models to extend the results to any vein geometry. The motivation behind the problem in this manuscript is interesting and meaningful. However, it has obvious expository and technical shortcomings, as detailed below:

1. In the manuscript, it is evident that there are significant issues with multiple images, such as irregular graphics, small fonts within the images, unclear image quality, and frequent occurrences of image inversion. Please comprehensively adjust and improve these images in the manuscript.

2. It can be observed from the manuscript that there is a notable absence of citations for figures and references, with a prominent lack of source attribution. Please thoroughly review and rectify these figure and reference citations in the manuscript.

3. It is evident from the manuscript that some sentences are incomplete, as seen in line 332 where crucial content is missing after the word "and," and there is a repetition of images. Please diligently address and rectify these issues in the manuscript.

4. In the manuscript, calibration of grid density, displacement rate, and plate-sample contact stiffness was conducted using LDB granite (Martin, 1993, and Labeid, 2019). Why was this rock chosen, which is completely different from the later CMET veined rock? Please provide a detailed explanation in the manuscript about its rationale and underlying considerations.

5. How is the implementation achieved as stated in the abstract, "This approach enables systematic investigation of any vein geometry without limitations of physical specimen availability or complexity of physical materials. Practical sample selection guidance is provided for planning geotechnical laboratory testing programs on rocks with single veins."? This study only employed two different types of rocks for parameter calibration, namely LDB granite (Martin, 1993, and Labeid, 2019) and CMET veined rock (de los Santos Valderrama, 2011), and the model verification section in the sixth part did not adequately present the comparison results with other literature. Please provide further supplementation and explanation in the manuscript, and incorporate additional comparison results with other literature to validate the model's worth.

6. In the conclusion section, the content appears relatively complex and cumbersome. It is suggested to moderately simplify and refine it in the manuscript. This can be achieved by summarizing and highlighting the main points while retaining key research findings and significant conclusions.

Comments on the Quality of English Language

No.

Reviewer 2 Report

Comments and Suggestions for Authors

The manuscript presents the original results of numerical finite element simulations performed to evaluate the effect of orientation and thickness of a single vein on the stiffness and strength of laboratory-scale specimens under uniaxial and triaxial compression. It has been shown that gradually increasing the complexity of numerical characteristics through a three-step iterative calibration allows one to achieve the desired calibration of the input properties in accordance with the target physical laboratory data of the UCS and TCS tests. The considered model was calibrated using laboratory data on non-veinlet Lac du Bonnet granite. The veined rock model was calibrated using experimental laboratory data of the mafic igneous complex (CMET) from the El Teniente mine, Chile.

The results of numerical simulation showed that the peak strength and fracture of rock volumes significantly depend on the properties of the material of the veins, their thickness and orientation relative to the direction of the acting load.

The manuscript is well structured.

Figures have good quality.

The list of references is satisfactory.

The conclusions are based on the analysis of the obtained numerical and experimental results.

The results presented in the manuscript can be considered as a scientific basis for solution of important problem related to the selection of a certain type of laboratory rock samples with veins for the calibration of a computational model for predicting the mechanical response of rocks with hydrothermal veins to loads.

The results presented in the manuscript may be of interest to a broad audience including specialists who study the influences of hydrothermal veins on contributing to load-driven rockmass failure.

The manuscript needs addition.

1)                 The manuscript should be supplemented with information about the software used in the finite element simulation.

2) It is necessary to substantiate the possibility to use data obtained during 2D modeling under plane strain conditions for calibrating models when experimental data obtained during testing of 3D samples.

3) It is necessary to supplement the manuscript with information about the loading conditions of the model volume, temporal dependence of applied displacement.

4) Although the manuscript provides information on used constitutive relations and criteria through reference section, direct formulas can raise the readability of the manuscript significantly.

Reviewer 3 Report

Comments and Suggestions for Authors

1.       Vein geometry varies widely in natural rock formations. Have you conducted any sensitivity analysis to evaluate how variations in vein geometry (orientation and thickness) might affect your model predictions? How robust are your conclusions across different ranges of vein orientations and thicknesses?

2.       Are there additional challenges in applying your approach to practical engineering design?

3.       Your conclusion briefly touches on factors like computational capacity and available time/budget for projects. Can you provide more details on the trade-offs between using discrete joint elements and equivalent continuum models in terms of computational requirements and accuracy? How might the choice of approach impact the outcomes of a real-world project?

4.       Are there specific areas where your approach could be extended or improved to address additional complexities?

5.       Could you elaborate on the relationship between the orientation of the vein and the decrease in Young's modulus for the whole specimen? How do variations in vein orientation impact the interaction between the vein and the core axis?

6.       You mentioned that the effect of vein thickness varies with respect to vein orientation. Can you provide more details on how these two factors interact? What's the underlying mechanism that leads to the different impact of thickness at different orientations?

7.       You mentioned that your study's findings partly agree with the Jaeger and Cook (1979) model. Could you discuss the implications of these agreements and disagreements? How might these differences affect the practical application of your study's findings?

8.       When discussing the potential increase in strength if the vein material were stronger than the wallrock, how does this scenario align with real-world geological characteristics? Are there instances where such a scenario is commonly encountered, and how does this impact practical applications?

9.       You mentioned that the Hoek-Brown strength envelopes are appropriate for ubiquitous and randomly oriented stockwork vein networks. Could you provide more context on how this characterization might differ from that of single veined specimens, and how this distinction might affect engineering applications?

10.   Why was the Mohr-Coulomb strength criterion chosen specifically for characterizing the shear strength of the intact veins? Were there specific advantages or limitations of this criterion that influenced its selection for your numerical modeling?

11.   The variation in shear strength envelope size based on different test configurations is interesting. Could you explain why the range of normal stresses is significantly larger when varying vein orientations compared to varying confining stress? How does this influence the accuracy and applicability of shear strength data?

Round 2

Reviewer 1 Report

Comments and Suggestions for Authors

All the comments have been addressed, and the reviewer suggested the manuscript to be accepted.

Comments on the Quality of English Language

All the comments have been addressed, and the reviewer suggested the manuscript to be accepted.

Reviewer 3 Report

Comments and Suggestions for Authors

The paper can be accepted.